# XQC: Well-conditioned Optimization Accelerates Deep Reinforcement Learning

**Daniel Palenicek**[1,2] **Florian Vogt**[3] **Joe Watson**[4] **Ingmar Posner**[4] **Jan Peters**[1,2,5,6]
[1]Technical University of Darmstadt [2]hessian.AI [3]University of Freiburg [4]University of Oxford
[5]German Research Center for AI (DFKI) [6]Robotics Institute Germany (RIG)
daniel.palenicek@tu-darmstadt.de

## ABSTRACT

Sample efficiency is a central property of effective deep reinforcement learning algorithms. Recent work has improved this through added complexity, such as larger models, exotic network architectures, and more complex algorithms, which are typically motivated purely by empirical performance. We take a more principled approach by focusing on the optimization landscape of the critic network. Using the eigenspectrum and condition number of the critic's Hessian, we systematically investigate the impact of common architectural design decisions on training dynamics. Our analysis reveals that a novel combination of batch normalization (BN), weight normalization (WN), and a distributional cross-entropy (CE) loss produces condition numbers orders of magnitude smaller than baselines. This combination also naturally bounds gradient norms, a property critical for maintaining a stable effective learning rate under non-stationary targets and bootstrapping. Based on these insights, we introduce XQC: a well-motivated, sample-efficient deep actor-critic algorithm built upon soft actor-critic that embodies these optimization-aware principles. We achieve state-of-the-art sample efficiency across 55 proprioception and 15 vision-based continuous control tasks, all while using significantly fewer parameters than competing methods. Our code is available at danielpalenicek.github.io/projects/xqc.

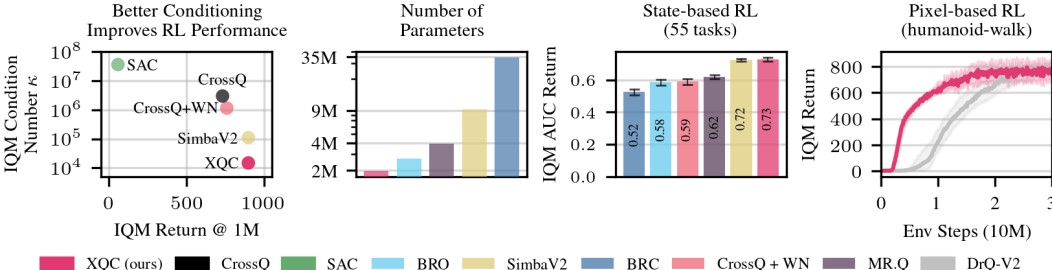

Figure 1: **Well-conditioned network architectures yield state-of-the-art RL performance.** Our algorithm, XQC with a BN and WN-based architecture and a CE loss, achieves competitive performance against state-of-the-art baselines across 55 proprioceptive continuous control tasks from four different benchmarks with a single set of hyperparameters. Notably, with $\sim 4.5\times$ fewer parameters and $\sim 5\times$ less compute in terms of FLOP/S than SIMBA-V2, the closest competitor. XQC's efficiency carries over to RL from pixels on 15 vision-based DMC tasks, significantly improving on DRQ-V2.

## 1 INTRODUCTION

Sample efficiency remains a major challenge in deep reinforcement learning (RL). Methods that can learn effectively from limited interactions are crucial for applying RL in domains such as robotics (Kober & Peters, 2008; Bohlinger et al., 2025), where generating data on real hardware is costly and time-consuming. Recent advances in model-free RL have primarily been driven by a paradigm of scaling—larger networks, higher update-to-data ratio (UTD) ratios, and ever-increasing computational budgets (Nikishin et al., 2022; Nauman et al., 2024; Lee et al., 2025a; Palenicek et al., 2025). These works view architectural improvements primarily as a means to scale up stably and in turn improve sample efficiency. While proven effective, this *'bigger is better'* paradigm comes at

the cost of computational efficiency and overlooks a more fundamental question: *Can we improve performance not by adding complexity, but by creating a better-conditioned optimization problem?*

To develop this principled understanding, we conduct a systematic investigation into three commonly used architectural components whose roles in RL are often guided by heuristics. First, we examine normalization layers. The RL community frequently uses layer normalisation (LN) (Ba et al., 2016), e.g., Ball et al. (2023), in contrast to batch normalisation (BN) (Ioffe & Szegedy, 2015), which until recently was thought to be problematic in the RL setting (Bhatt et al., 2024). Second, we consider weight normalisation (WN) (Lyle et al., 2024; Loshchilov et al., 2025; Palenicek et al., 2025) by periodically projecting the network's weights to the unit sphere, permitted through the normalization layers' *scale invariance* property. A technique known to improve the effective learning rate (ELR) (Van Laarhoven, 2017). Lastly, we study the critic's loss function. Distributional critics using a categorical cross entropy (CE) loss have grown in popularity (Bellemare et al., 2017). The conventional argument for their adoption is that modeling the full distribution of returns provides a better learning signal compared to mean squared error (MSE) regression (Farebrother et al., 2024); there is evidence that this loss is easier to optimize (Imani & White, 2018).

Through a systematic eigenvalue analysis of the critic's Hessian, we provide a principled explanation for *why* different architectures outperform others. Our analysis first shows that BN consistently produces better-conditioned local loss landscapes than LN during learning, with condition numbers that are orders of magnitude smaller. Second, our investigation of the critic loss reveals that, beyond its representational advantages, the CE loss induces a remarkably well-conditioned optimization landscape compared to the MSE loss. We find that the combination of BN, WN, and a categorical CE loss works in synergy to dramatically improve the conditioning of the optimization problem and stabilize the ELR, a key metric for maintaining plasticity in deep RL. In summary, we claim the following contributions:

1. XQC, a simple and efficient extension to soft actor critic, uses the powerful synergy between BN, WN, and a distributional critic with a CE Bellman error loss for sample-efficient learning.

2. A Hessian eigenvalue analysis of modern deep RL critics, revealing the superior conditioning properties of distributional critic losses over the mean squared error.

3. Extensive empirical validation on 55 proprioception and 15 vision-based continuous control tasks, demonstrating state-of-the-art performance against more complex, larger-scale methods.

## 2 PRELIMINARIES

This section briefly introduces the necessary background and notation for this paper.

**Deep reinforcement learning.** In this work, we assume the standard RL setting (Sutton & Barto, 2018), where an agent attempts to learn a policy that maximizes its expected discounted return. Our experiments are based on the popular off-policy actor-critic algorithm soft actor-critic (SAC) (Haarnoja et al., 2018), where policy and critic are represented by neural networks. A key quantity in reinforcement learning with function approximation is the Bellman error $\Delta_{\boldsymbol{\theta}}$,

$$\Delta_{\boldsymbol{\theta}}(\boldsymbol{s}, \boldsymbol{a}) = Q(\boldsymbol{s}, \boldsymbol{a}) - Q_{\boldsymbol{\theta}}(\boldsymbol{s}, \boldsymbol{a}), \quad Q(\boldsymbol{s}, \boldsymbol{a}) = r(\boldsymbol{s}, \boldsymbol{a}) + \gamma \, \mathbb{E}_{\boldsymbol{s}' \sim p(\cdot | \boldsymbol{s}, \boldsymbol{a})}[V(\boldsymbol{s}')], \tag{1}$$

where $Q$ and $V$ are the 'soft' parametric critic and value functions, respectively. Minimizing this Bellman error effectively is key to the success of actor-critic methods (Sutton & Barto, 2018). This error is typically minimized with the mean squared regression loss and gradient-based optimization. The distributional C51 algorithm (Bellemare et al., 2017) reformulates the task as a classification problem. Instead of a scalar estimate, the function approximator outputs the logits of a categorical distribution over the full support of $Q$. The Bellman error can then be minimized using a categorical CE loss, which has been shown to improve performance and stability (Farebrother et al., 2024).

**Analyzing gradient-based optimization.** To analyze the optimization of our gradient-based updates, we consider their first- and second-order aspects. For gradient-based optimization with parameter normalization, we must consider the effective learning rate (Definition 1).

**Definition 1.** *(Effective learning rate, ELR, Van Laarhoven (2017)). For a scale-invariant function $f(\boldsymbol{\theta}) = f(\lambda \boldsymbol{\theta}), \lambda > 0$, the 'effective' learning rate $\tilde{\eta}$ for an update $f(\boldsymbol{\theta} + \eta \, \boldsymbol{g}(\boldsymbol{\theta}))$ with gradients $\boldsymbol{g}(\boldsymbol{\theta})$ is the learning rate when taking this scale invariance into account,*

$$f(\boldsymbol{\theta} + \eta \, \boldsymbol{g}(\boldsymbol{\theta})) = f(\tilde{\boldsymbol{\theta}} + \tilde{\eta} \, \boldsymbol{g}(\tilde{\boldsymbol{\theta}})), \quad \tilde{\eta} = \eta / \|\boldsymbol{\theta}\|_2, \quad \tilde{\boldsymbol{\theta}} = \boldsymbol{\theta} / \|\boldsymbol{\theta}\|_2.$$

Figure 2: When performing gradient-based optimization, the condition number ($\kappa$) of the objective's Hessian significantly impacts convergence. We illustrate this phenomenon with a simple two-dimensional quadratic example. As $\kappa$ increases by an order of magnitude, gradient descent converges at a lower rate. We believe this phenomenon plays a similar role when learning the critic in deep reinforcement learning, where high condition numbers lead to poor sample efficiency.

Recent work has studied ELR in the context of loss of 'plasticity' in neural networks and scaling gracefully to larger UTD ratios in RL (Lyle et al., 2024; Palenicek et al., 2025).

To analyze the second-order properties of the loss landscape, a local quadratic approximation

$$\mathcal{L}(\boldsymbol{\theta} + \delta\boldsymbol{\theta}) \approx \mathcal{L}(\boldsymbol{\theta}) + \nabla_{\boldsymbol{\theta}}\mathcal{L}(\boldsymbol{\theta})\delta\boldsymbol{\theta} + \frac{1}{2}\delta\boldsymbol{\theta}^{\top}\nabla_{\boldsymbol{\theta}}^{2}\mathcal{L}(\boldsymbol{\theta})\delta\boldsymbol{\theta}, \tag{2}$$

illustrates the role of the Hessian $\nabla_{\boldsymbol{\theta}}^{2}\mathcal{L}(\boldsymbol{\theta})$ in characterizing the curvature of the local loss landscape, which is measured using its eigenvalues. The $\beta$-smoothness of the objective upper-bounds the largest eigenvalue of the Hessian (Definition 2), while the ratio of largest to smallest absolute values describes the condition number (Definition 3). The larger the condition number, the less effective gradient descent with a fixed learning rate will be due to the large range in curvature per dimension, as illustrated in Figure 2 (Nocedal & Wright, 2006). While we use an adaptive learning rate optimizer (Adam, Kingma & Ba (2015)), whose adaptivity helps overcome issues with ill-conditioning, the loss landscape curvature remains relevant when assessing optimization difficulty.

**Definition 2.** *($\beta$-smoothness, Aravkin et al. (2017)). A loss function $\mathcal{L}(\boldsymbol{\theta})$ is said to be $\beta$-smooth if its gradient is Lipschitz continuous with constant $\beta$, i.e., $\|\nabla_{\boldsymbol{\theta}}\mathcal{L}(\boldsymbol{\theta}_1) - \nabla_{\boldsymbol{\theta}}\mathcal{L}(\boldsymbol{\theta}_2)\| \leq \beta \|\boldsymbol{\theta}_1 - \boldsymbol{\theta}_2\|$ which is equivalent to the largest eigenvalue of its Hessian being bounded by $\beta$, i.e., $\lambda_{\max}(\nabla_{\boldsymbol{\theta}}^{2}\mathcal{L}(\boldsymbol{\theta})) \leq \beta$. As such, the $\beta$-smoothness quantifies the maximum curvature of the landscape.*

In our experiments, we will look at the largest eigenvalue as a proxy for the empirical measure of $\beta$.

**Definition 3.** *(Condition number, Nocedal & Wright (2006)). For a normal matrix $\boldsymbol{H} \in \mathbb{R}^{d \times d}$ with eigenvalues $\lambda_1, \ldots, \lambda_d$, its condition number $\kappa$ is $\kappa(\boldsymbol{H}) = \max_i |\lambda_i| / \min_i |\lambda_i|$. As a measure of sensitivity, a low condition number describes a 'well-conditioned' matrix, while a high condition number describes an 'ill-conditioned' matrix.*

For a Hessian, the condition number is used to analyze and characterize the effectiveness of gradient-based optimization algorithms (Nocedal & Wright, 2006). Using the insights of the effective learning rate, $\beta$-smoothness, and condition numbers, we now compare the optimization landscapes of different actor-critic architectures in deep RL.

## 3 THE OPTIMIZATION LANDSCAPES OF THE BELLMAN ERROR

We seek to improve sample efficiency by enhancing the critic's optimization landscape. This section applies the optimization insights from Section 2 to the Bellman error minimization. Hessian eigenvalues have previously been used to understand the benefits of batch normalization in supervised learning (Ghorbani et al., 2019). To our knowledge, this is the first such analysis for deep RL.

### 3.1 AN EMPIRICAL INVESTIGATION OF CRITIC OPTIMIZATION.

To quantify the impact of common architectural components on the optimization of the Bellman error, we looked at the eigenvalues of the critic's Hessian while learning the challenging DeepMind control suite (DMC) `dog-trot` environment, a high-dimensional continuous control task, ensuring the findings are not artifacts of a trivial toy-task.

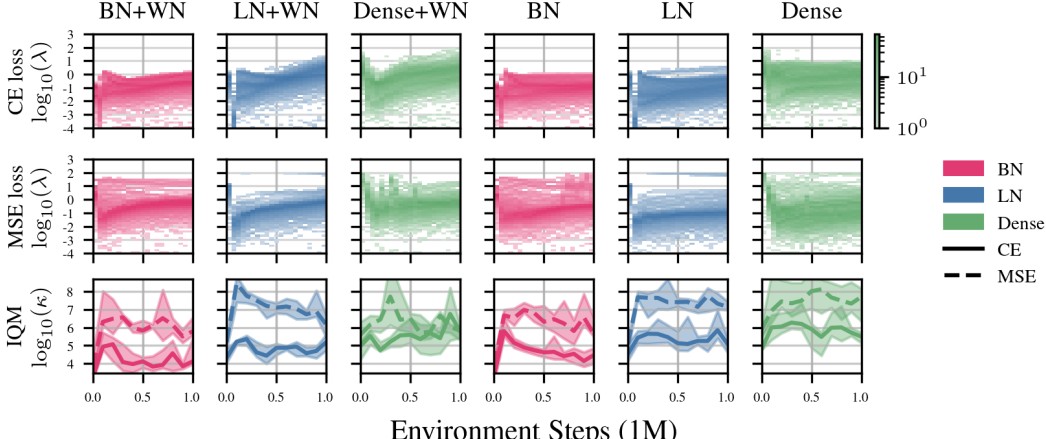

Figure 3: Eigenvalues and condition numbers on `dog-trot` over 5 seeds for different critic architectures during training. The top and middle rows show the eigenspectra of the CE loss and MSE loss, respectively. The columns correspond to different combinations of normalization layers and WN. The bottom row shows the IQM and 90% SBCI of the condition number $\kappa$ aggregated over five seeds for CE and MSE losses, respectively. Architectures using BN show more compact and stable eigenspectra over the course of training with no outliers. LN suffers from large outlier modes and includes overall larger eigenvalues. Similarly, the CE loss significantly improves loss landscape conditioning over an MSE.

We systematically compare critic networks with combinations of common architectural components:

- Normalization strategies: BN, LN, None (Dense).
- Weight projection to the unit sphere: WN (□), no WN (■).
- Loss functions: MSE (△), CE (○).

This results in a total of 12 distinct architectural combinations, allowing for a thorough dissection of each component's contribution. Per architecture, we run 5 random seeds for 1M environment steps and compute the Hessian eigenspectrum at 20 checkpoints throughout training using an efficient JAX (Bradbury et al., 2018) implementation of the *stochastic Lanczos quadrature* algorithm (Golub & Welsch, 1969; Lin et al., 2016), adapted from Ghorbani et al. (2019).

**Eigenvalue analysis.** First, we qualitatively analyze how the eigenvalues evolve during training for the different architectures. Figure 3 (top & middle) reveals striking differences in the curvature of the loss landscape for the different components. Architectures employing BN consistently produce more compact and stable eigenspectra throughout training, with eigenvalues remaining bounded within a moderate range and free of significant outliers. In stark contrast, LN architectures suffer from large, growing outlier eigenvalues, signifying sharp curvature that can destabilize training. Similarly, the CE loss significantly improves loss landscape conditioning over an MSE. This is also reflected in the condition numbers Figure 3 (bottom), where BN-based architectures are consistently an order of magnitude smaller and more stable than their non-BN counterparts.

**Condition numbers and $\beta$-smoothness.** To make the relationship between the spectral properties and performance explicit, Figure 4 presents the data in an aggregated form. Each point shows aggregated results over 5 seeds, correlating an architecture's IQM condition number, IQM $\max(\lambda)$ and IQM Kurtosis$(\lambda)$, respectively, over the entire course of training, with its sample-efficiency (IQM return at 1M timesteps). These plots show a clear and strong trend: architectures with lower condition numbers and smaller maximum eigenvalues achieve higher returns. This trend provides compelling empirical evidence that, perhaps unsurprisingly, a smoother, better-conditioned optimization landscape is a key driver of performance in deep RL. The Kurtosis provides a proxy measure for outliers in the eigenspectrum, where BN-based architectures consistently show lower Kurtosis than their LN-based counterparts. Furthermore, the results show that BN, WN, and a categorical CE loss each independently improve the landscape's conditioning, and when combined, their synergistic effect yields the best-conditioned landscape and the highest performance.

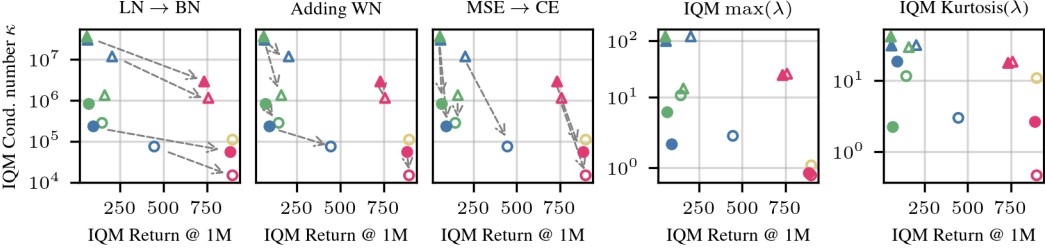

Figure 4: The condition numbers and maximum eigenvalues against the return at 1M steps on DMC dog-trot. Normalization strategies are color-coded BN, LN, Dense. Use of WN = empty shape □, whereas no WN is represented by a filled shape ■. MSE loss = △ and CE = ○. Architectures with lower condition numbers and lower maximum eigenvalues tend to have better final returns. Also, BN, WN, and the categorical CE loss each improve the loss conditioning independently (columns 1-3). Combined, they result in the best conditioning and best performance ○. For reference, we include SIMBA-V2 ○ a strong baseline with a similarly low condition number.

## 3.2 WHY DOES CROSS-ENTROPY OUTPERFORM THE SQUARED ERROR?

Distributional RL and C51 were proposed to perform distributional regression of the returns, beyond predicting only the average value (Bellemare et al., 2023). In this section, we motivate distributional losses from the optimization perspective (Imani & White, 2018) to explain the dramatic difference in condition numbers between CE and MSE Bellman errors in Section 3.1. We show that the CE loss has desirable properties for optimization over the MSE. Firstly, Propositions 1 and 2 show that the gradient norm for the loss with respect to the predictions can only be bounded for the CE loss.

**Proposition 1.** *The loss, $l(\boldsymbol{y}, \hat{\boldsymbol{y}}) = \frac{1}{2}||\boldsymbol{y} - \hat{\boldsymbol{y}}||_2^2$ has unbounded gradients w.r.t. $\hat{\boldsymbol{y}}$,*

$$||\nabla_{\hat{\boldsymbol{y}}} l(\boldsymbol{y}, \hat{\boldsymbol{y}})||_2 = ||\boldsymbol{y} - \hat{\boldsymbol{y}}||_2 \leq \infty, \quad \hat{\boldsymbol{y}} = \boldsymbol{f}_{\boldsymbol{\theta}}(\boldsymbol{x}). \tag{3}$$

**Proposition 2.** *The loss, $l(\boldsymbol{t}, \hat{\boldsymbol{y}}) = -\sum_{i=1}^{d} t_i \log \hat{t}_i$, $\hat{\boldsymbol{t}} = Softmax(\hat{\boldsymbol{y}})$ has bounded gradients w.r.t. $\hat{\boldsymbol{y}}$,*

$$||\nabla_{\hat{\boldsymbol{y}}} l(\boldsymbol{t}, \hat{\boldsymbol{y}})||_2 = ||\boldsymbol{t} - Softmax(\hat{\boldsymbol{y}})||_2 \leq \sqrt{2}, \quad \hat{\boldsymbol{y}} = \boldsymbol{f}_{\boldsymbol{\theta}}(\boldsymbol{x})). \tag{4}$$

Combining Proposition 2 with weight normalization and a Lipschitz assumption, we can upper bound the effective gradient update (Definition 1) in Theorem 1.

**Theorem 1.** *For the cross entropy-loss $l$ and learning rate $\eta \geq 0$, for a scale-invariant function approximator $\boldsymbol{f}_{\boldsymbol{\theta}}$ which is $L_{\boldsymbol{f}}$ Lipschitz continuous in the L2 norm with respect to $\boldsymbol{\theta}$ with fixed parameter norm $||\boldsymbol{\theta}||_2 = C$, the effective gradient update can be upper bounded as*

$$\eta ||\boldsymbol{\theta}||_2^{-1} \nabla_{\boldsymbol{\theta}} l(\boldsymbol{t}, \boldsymbol{f}_{\boldsymbol{\theta}}(\boldsymbol{x})) \leq \eta C^{-1} \sqrt{2} L_{\boldsymbol{f}}. \tag{5}$$

To analyze second-order properties, we must assume that we can bound the eigenvalues of the function approximator's Hessian so that weight decay can ensure the Hessian of the objective is positive definite and the smallest eigenvalue is greater than zero.

**Assumption 1.** *We assume eigenvalue bounds for the function approximator Hessian (per output),*

$$0 \leq |\lambda_1^f| \leq |\sigma_i(\nabla_{\boldsymbol{\theta}}^2 f_{\boldsymbol{\theta}}(\boldsymbol{x}))| \leq |\lambda_m^f| < \infty \quad \forall i \in [0, m], \boldsymbol{x} \in \mathcal{X}, \boldsymbol{\theta} \in \Theta.$$

**Proposition 3.** *Given Assumption 1, the eigenvalues of the Hessian of the mean squared error loss with weight decay $\mu^2$, $\mu \geq 0$ are unbounded and the condition number cannot be upper bounded.*

**Proposition 4.** *Given Assumption 1, the eigenvalues of the Hessian of the cross-entropy loss with weight decay $\mu^2$ have an upper-bounded condition number*

$$\kappa\left(\nabla_{\boldsymbol{\theta}}^2 \mathcal{L}\right) \leq (4\lambda_m^f + L_{\boldsymbol{f}}^2 + \epsilon)/\epsilon, \quad \epsilon \geq 0,$$

*when $\mu^2 = 2\lambda_m^f + \epsilon$, which provides a finite upper bound when $\epsilon > 0$.*

For proofs see Appendix J. In practice, we do not require weight decay to attain good performance, and as a result, the Hessian was observed to not always be positive definite. Nonetheless, these results provide a formal intuition for our empirical result that CE losses consistently report smaller condition numbers. Our analysis directly motivates the design of our XQC algorithm, presented next.

## 4 XQC: A SIMPLE & WELL-CONDITIONED ACTOR-CRITIC ARCHITECTURE

This section presents our novel XQC algorithm, a direct conclusion of our optimization analysis in Section 3. It is a simple, yet powerful architecture with the purpose of improving the loss landscapes optimization behaviours, that extends the popular SAC algorithm. XQC's critic architecture is motivated by a central principle: combining components that synergistically improve optimization dynamics. We provide a complete list of hyperparameters in Appendix A.

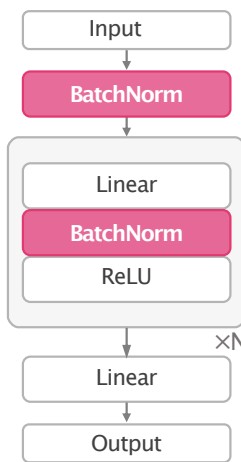

**Batch normalization.** XQC uses BN layers directly on the network input and after each linear layer (Figure 5). Following Bhatt et al. (2024), we implement a joined forward pass to automatically calculate the BN running statistics on the joined $(s, a)$ and $(s', a')$ distribution (Bhatt et al., 2024), to successfully integrate BN in the RL loop. In contrast to Bhatt et al. (2024), we find that switching the order of normalization and ReLU-activation leads to better performance. It has the added benefit that in this order, BN's scale invariance is preserved for any activation function, as opposed to homogeneous ones only. XQC uses four hidden layers with 512 neurons each.

**Cross-entropy Bellman loss.** We use a C51-style categorical critic with 101 atoms and a CE loss (Bellemare et al., 2017). We use standard reward normalization based on running statistics of the standard deviation of the return $R$ to effectively bound the $Q$ values to the support of our categorical critic $\hat{r}_t = r_t/\sigma(R)$ (Engstrom et al., 2020). Next to improving the loss landscape conditioning, another desirable property of the categorical CE loss is to keep gradient norms bounded and thereby help keep the ELR constant.

Figure 5: The XQC network architecture consists of only three standard components: Linear, BN, and ReLU for a total of 4 hidden layers.

**Weight normalization.** Enabled by BN's scale-invariance property, we project the weights of each dense layer to the unit sphere after each gradient step. This normalization keeps the denominator of the ELR constant, so it becomes practically constant when using the CE loss (Figure 8), so XQC maintains good plasticity. With a constant ELR, we can now leverage a learning rate schedule for Adam (Kingma & Ba, 2015) as previously suggested (Lyle et al., 2024; Lee et al., 2025b).

**Vision encoder.** For experiments on the DMC vision-based environments, we use the standard DRQ-V2 (Yarats et al., 2022) image encoder. For a fair and direct comparison to DRQ-V2, we use its standard, unmodified vision encoder, which consists of convolutional layers alternated with ReLU activations, followed by a linear layer, LN, and a tanh activation. Our architectural modifications are confined to the subsequent MLP layers of the actor and critic.

## 5 EXPERIMENTS

This section empirically validates our central hypothesis: that the synergistic combination of BN, WN, and a categorical CE critic loss, designed to create a well-conditioned optimization landscape, directly translates into state-of-the-art sample efficiency and training stability. We structure our experiments to first demonstrate XQC's superior performance against strong baselines (Section 5.1), then dissect the underlying mechanics through analysis of common plasticity metrics and the ELR (Section 5.2), Finally, in Section 5.3 we analyze computational efficiency, scaling properties, and present a thorough ablation study (Appendix G) confirms the necessity of each of XQC's architectural components.

**Evaluation metrics.** For the main experiments we run 10 random seeds per environment for 1 million environment steps and for ablations 5 seeds, unless otherwise noted. For statistically rigorous evaluations, we report the IQM and 90% SBCI for all aggregate scores, following the recommended best practices of Agarwal et al. (2021). To aggregate IQM return curves over multiple environments and benchmarks, each score needs to be normalized. We follow standard practice, details in Appendix B. In aggregated bar charts, we present *area under the curve* (AUC) of the IQM normalized return curve. The AUC captures both training speed as well as absolute performance simultaneously.

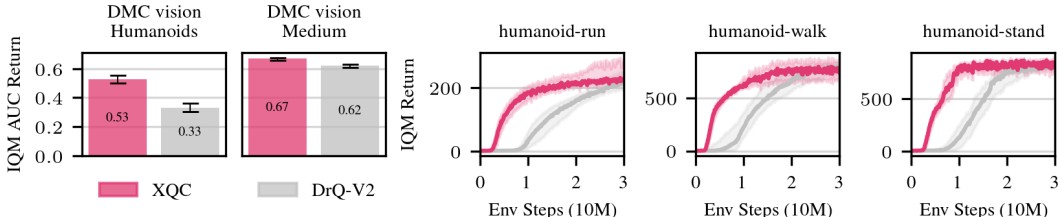

Figure 6: **XQC achieves state-of-the-art sample efficiency across 55 proprioceptive continuous control tasks.** We report the IQM AUC of normalized returns. Error bars denote 90% SBCIs. The right column shows total aggregated performance across the benchmarks (55 tasks). XQC matches or outperforms strong baselines, especially on the hardest DMC and HB tasks, while using a simpler and smaller architecture (see Section 5.3).

Figure 7: **XQC improves sample efficiency on 15 vision-based DMC tasks.** The left two columns show aggregated IQM AUC, demonstrating a significant performance advantage over the strong DRQ-V2 baseline, particularly on the difficult humanoid tasks. The right three columns show full training curves for the three humanoid environments (IQM over 10 seeds), highlighting XQC's significantly better sample-efficiency. For these experiments, XQC uses the standard DRQ-V2 encoder and hyperparameters, isolating performance gains to our proposed well-conditioned critic architecture.

As such, it discriminates between two algorithms, which converge to the same performance but at different speeds, as opposed to the IQM of the final performance.

**Benchmarks.** To validate XQC's effectiveness, we conduct comprehensive experiments across 70 continuous control tasks spanning five popular benchmark suites. Our evaluation covers 15 vision-based tasks from DMC, plus and additional 55 proprioceptive tasks from HumanoidBench (HB) (Sfer-razza et al., 2024) (14 tasks), DMC (Tassa et al., 2018) (25 tasks), MyoSuite (Myo) (Caggiano et al., 2022) (10 tasks), and MuJoCo (Todorov et al., 2012) (6 tasks). Our extensive evaluation shows XQC's generality, using a single set of hyperparameters across all tasks.

**Baselines.** We compare to several strong, recent model-free baselines: SIMBA-V2 (Lee et al., 2025b), BRO (Nauman et al., 2024), MRQ (Fujimoto et al., 2025), BRC (Nauman et al., 2025), CROSSQ+WN (Palenicek et al., 2025), SAC (Haarnoja et al., 2018), When available, we use the respective authors' evaluation results; otherwise, we run experiments using their official open-source implementations. Full details on baseline results are provided in Appendix C.

## 5.1 SAMPLE EFFICIENCY RESULTS

We start our experiments by investigating the training performance in terms of sample efficiency and comparing it to state-of-the-art baselines. All of these results use 10 seeds per environment. First, we present the proprioception-based results and then the vision-based tasks.

**Reinforcement learning from proprioception.** Figure 6 shows that XQC matches our outperforms strong baselines SIMBA-V2, MRQ, BRO and CROSSQ+WN on all 4 benchmarks. The rightmost column shows that on average XQC performs as well SIMBA-V2 while using significantly less network parameters and a substantially simpler architecture Figure 5. Notably XQC shows exceptional performance on the most complex tasks HB and DMC-hard. These enm induce notoriously difficult and ill-conditioned optimization landscapes. XQC's superior performance and learning speed suggest that its well-conditioned critic—characterized by a stable ELR and bounded gradients as shown in Section 5.2—is fundamentally better equipped to handle the non-stationary targets and bootstrapping errors inherent in these challenging domains.

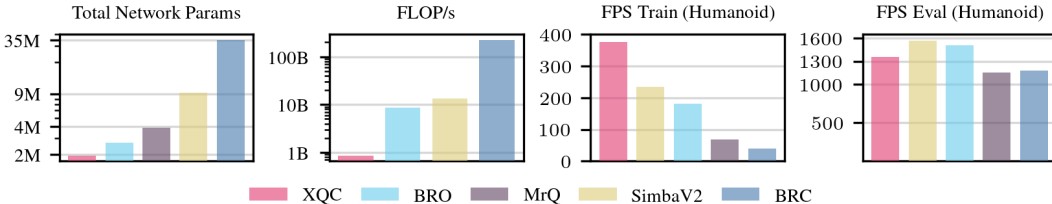

Figure 8: **XQC's architecture creates exceptionally stable learning dynamics.** For XQC, BN +WN stabilizes the parameter norm, while BN +CE keep the gradient norm and ELR near constant.

**Reinforcement learning from pixels.** On the vision-based DMC environments, we compare XQC to DRQ-V2. For these results, we re-implemented XQC in the official DRQ-V2 codebase. We used the DRQ-V2 encoder and the same hyperparameters as the original DRQ-V2 to make the comparison as fair as possible. Figure 7 shows that on most tasks, XQC outperforms or at least matches DRQ-V2 performance. This is most pronounced on the much more challenging humanoid tasks. Learning the vision-encoder from scratch requires a large number of samples in itself. We hypothesize that this is why the performance increase of XQC is smaller on the easier tasks and much more pronounced on the humanoids, which have a $10\times$ overall runtime.

## 5.2 PLASTICITY ANALYSIS

Analysing the improvement of common plasticity metric confirms the effectiveness of XQCs design principles. Figure 8 presents plasticity metrics aggregated over all 55 proprioceptive tasks. XQC w/o WN's growing parameter norms decrease the ELR towards zero over time, reconfirming the findings of Lyle et al. (2024) and Palenicek et al. (2025). We notice that the ELR appears directly coupled to the *gradient norm*, for all architectures employing WN. While XQC MSE controls the parameter norm, its gradients are unbounded and heavily influenced by outliers; consequently, its gradient norm and ELR grow over the course of training by about one order of magnitude (requiring a second y-axis to compare). XQC's CE critic loss removes this disturbance, directly reflected in remarkably stable gradient norm and ELR throughout the course of training, which are many orders of magnitude smaller. We show per benchmark plasticity metrics in Appendix F.

## 5.3 PARAMETER AND COMPUTE EFFCIENCY

XQC achieves competitive sample efficiency while requiring $\sim 4.5\times$ fewer parameters than SIMBA-V2 (Figure 9). This parameter efficiency directly results in high computational efficiency with $\sim 5\times$ fewer FLOP/S than SIMBA-V2 and BRO and $> 100\times$ fewer FLOP/S than BRC. We conjecture that XQC's superior computational efficiency is rooted in its well-conditioned architecture. With respect to practical performance, XQC shows $60\%$ higher frames-per-second (FPS) during training, measured on the MuJoCo humanoid on a NVIDIA RTX 4090 workstation, resulting in significantly faster wall-clock times. XQC's evaluation performance is measured at $1355$FPS, slightly lower the SIMBA-V2 baseline due to its slightly larger policy networks, but remains more than fast enough typical control frequencies seen in deep reinforcement learning (10–50 Hz).

Figure 9: **XQC is significantly more parameter and compute efficient than competitive baselines.** $\sim 4.5\times$ fewer parameters and $\sim 5\times$ fewer FLOP/S than SIMBA-V2 and BRO and $> 100\times$ fewer FLOP/S than BRC. The computational efficiency translates into a significantly higher training FPS measured on the MuJoCo humanoid environment on an NVIDIA RTX 4090 workstation.

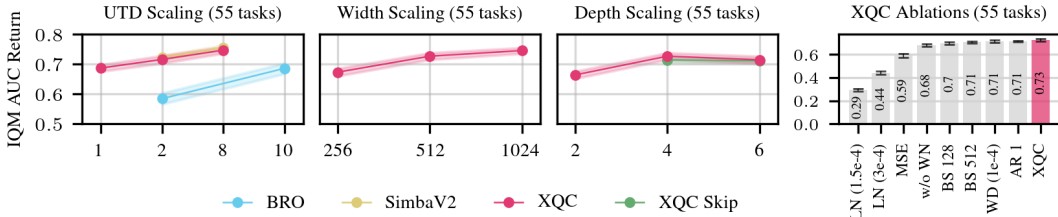

Figure 10: **XQC scales stably with UTD, network depths, and widths.** Columns 1–3: XQC's scaling in terms of UTD, layer width, and layer depth. Increasing compute and model capacity generally improves or maintains performance, demonstrating that our well-conditioned design is robust to scaling. Results are aggregated across all 55 proprioception tasks and 5 seeds each. Right column: We perform multiple architectural and parameter ablations with respect to the full XQC algorithm. XQC is robust to changing the action repeat (AR=1), introducing weight decay (WD), and different batch sizes (BS∈ {128, 512}). We further ablate all three of XQC's main components, first without WN (w/o WN), second replacing the CE loss with an MSE loss (MSE), and third replacing BN with LN. Each component's removal results in a significant performance drop, showing their synergistic contribution.

## 5.4 XQC SCALING BEHAVIOUR AND ARCHITECTURE ABLATIONS

Figure 10 colums 1–3 demonstrate XQC's scaling ability. XQC improves performance with increasing UTD at a similar slope to SIMBA-V2, with a fraction of the parameters (Section 5.3). Similarly, XQC improves or maintains performance with larger and deeper networks. These results demonstrate robustness towards different hyperparameters and XQC's ability to scale stably, enabled by its well-conditioned architectural design. This property is desirable since increasing compute always represents a trade-off between sample efficiency and wall-clock or energy. It allows practitioners to use their available budget most efficiently. For depth scaling, we also ablate the influence of skip connections but find no significant effect for the investigated depths. Additionally, we investigate the influence of different batch sizes ∈ {128, 512} compared to the default of 256, the use of weight decay, and the impact of action repetition. Neither of these ablations shows a significant influence on XQC. Appendix I presents additional conditioning experiments for the tested components. To demonstrate the synergy between the proposed architectural components, we ablate each of the three main components of the proposed XQC architecture: BN, WN, and the CE distributional critic. Results for all 55 and 5 seeds are shown in Figure 10 with per benchmark ablations in Figure 22. Our analysis confirms that each of the components is vital, especially in the most difficult DMC-hard and HB suits. The largest drop in performance occurs when switching BN for LN. We further investigate LN with half the learning rate, accounting for the increased ELR of LN (See Figure 8); however, this leads to even worse performance. This is in line with our finding that LN-based architectures show lower performance due to worse conditioning. While using a MSE critic loss has the second most significant influence, the removal of WN is still substantial, but shows the lowest overall impact. In summary, combining each of the three components is vital for XQC's performance.

## 6 RELATED WORK

Our work is positioned at the intersection of several key research areas in deep RL that all attempt to improve sample efficiency. A prevailing trend for improving sample efficiency has been scaling along different axes. *Compute scaling*, particularly increasing the UTD ratio, has recently been a major focus. However, naively increasing UTD can lead to instability and overfitting on early experience (Nikishin et al., 2022). Researchers have suggested many different regularizers to stably increase the UTD; From full parameter-resets (Nikishin et al., 2022; D'Oro et al., 2022), to critic ensembles (Chen et al., 2021) and drop-out (Hiraoka et al., 2021), to normalization layers like LN (Hiraoka et al., 2021; Lyle et al., 2024; Nauman et al., 2024; Lee et al., 2025a), BN (Bhatt et al., 2024), hyper-spherical normalization (Hussing et al., 2024; Lee et al., 2025b) and spectral normalization (Bjorck et al., 2021). Recently, works have found that the combination of normalization layers together with WN can stabilize the ELR, helping against loss of plasticity (Lyle et al., 2024) and also enable scaling RL to high UTD ratios (Palenicek et al., 2025). Recent works have worked

on developing scaling laws for RL (Rybkin et al., 2025; Fu et al., 2025). *Network scaling*, is another path authors are exploring to increase sample efficiency. Bhatt et al. (2024) showed that BN allowed them to significantly scale the layer width. Since then, authors have looked into specific LN-based architectures (Nauman et al., 2024; Lee et al., 2025a) and network sparsity (Ma et al., 2025). Lee et al. (2025a) propose a 'simplicity bias' score, computed using an FFT and scoring 'simplicity' higher for functions with lower frequency content across random initializations. This score has no theoretical justification relating it to sample efficiency or generalization. Another line of research attempts to scale network sizes dynamically during training (Liu et al., 2025; Kang et al., 2025). Concurrent work Castanyer et al. (2025) combine second-order optimization and multi-skip residual connections to improve scaling and monitor the trace of the Hessian for deep value-based RL. Orthogonal to our work, recent methods revisit the target network design itself. Vincent et al. (2026) share features between online and target networks to reduce memory while preserving stability, and Hendawy et al. (2026) take the minimum of both estimates for bootstrapping to accelerate learning. *Model-based* methods scale computation by learning a separate dynamics model, which is leveraged in the RL loop (Janner et al., 2019; Hafner et al., 2020; Palenicek et al., 2023; Hansen et al., 2024).

## 7    CONCLUSION & FUTURE WORK

In this work, we shifted the focus from the prevailing pure scaling goal in deep RL and instead focus on improving the critic's optimization landscape. Through an eigenvalue analysis of the critic's Hessian, we demonstrate that specific architectural choices, namely batch normalization, weight normalization, and a distributional cross-entropy loss, create a better optimization landscape with a condition number orders of magnitude smaller during learning. This superior conditioning translates directly into learning performance gains. We propose XQC, an algorithm embodying these principles, which achieves state-of-the-art sample efficiency across 70 continuous control tasks from proprioception and vision domains. XQC accomplishes this performance with significantly fewer parameters than competing methods, underscoring that a principled focus on optimization fundamentals can yield greater performance and efficiency than brute-force scaling alone. For future work, insights from XQC can be used to accelerate actor-critic model-based and imitation learning algorithms, e.g., Janner et al. (2019), Watson et al. (2023). XQC performance could be further accelerated by incorporating prior datasets, e.g., Ball et al. (2023).

## ACKNOWLEDGEMENTS

This research was funded by the research cluster "Third Wave of AI", funded by the excellence program of the Hessian Ministry of Higher Education, Science, Research and the Arts, hessian.AI and by the Deutsche Forschungsgemeinschaft (DFG, German Research Foundation) under Germany's Excellence Strategy (EXC-3057/1 "Reasonable Artificial Intelligence", Project No. 533677015). It was further supported by a UKRI/EPSRC Programme Grant [EP/V000748/1] and partially supported by the German Federal Ministry of Research, Technology and Space (BMFTR) under the Robotics Institute Germany (RIG).

## REPRODUCIBILITY STATEMENT

We took special care to ensure this work is reproducible and will make the code open source upon acceptance. To ease reproducibility, algorithm details are explained Section 4, all hyperparameters are listed in Appendix A, and training curves are shown Appendices D and E.

## LARGE LANGUAGE MODEL USAGE

A large language model was helpful in polishing writing, improving reading flow, and identifying remaining typos.

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

## A    Hyperparameters

Table 1 summarizes all proprioception-based experiments' hyperparameters.

$$\gamma = \text{clip}\left(\frac{\frac{T}{5} - 1}{\frac{T}{5}}, [0.95, 0.995]\right),$$

where $T$ denotes the effective episode length, calculated by dividing the episode length by the number of repeated actions. We use the default hyperparameters used in their respective GitHub repositories for all other baselines. One exception is BRC, where we reduced the number of parameters from the default 256M to 64M. As noted by the authors, using more than 64M parameters does not provide additional benefit in the single-task setting, which is the focus of our work.

Table 2 contains all hyperparameters for the vision-based experiments based on the official DRQ-V2 codebase.

Table 1: Hyperparameters for XQC and baselines on all proprioception tasks.

| Hyperparameter | XQC | CROSSQ+WN | SAC | SIMBA-V2 | BRO | BRC | MRQ |
|---|---|---|---|---|---|---|---|
| Block design | Dense(dim) BN ReLU | Dense(dim) ReLU BN | Dense(dim) ReLU | Dense(4 × dim) Scaler ReLU Dense(dim) L2 Norm LERP L2 Norm | Dense(dim) LN ReLU Dense LN Skip Connection | Dense(dim) LN ReLU Dense LN Skip Connection | Dense(dim) LN Activation |
| Critic learning rate | 0.0003 | 0.0003 | 0.0003 | 0.0001 | 0.0003 | 0.0003 | 0.0003 |
| Critic hidden dim | 512 | 512 | 256 | 512 | 512 | 2048 | 512 |
| Critic number of blocks | 4 | 2 | 2 | 2 | 2 | 2 | 3 |
| Actor learning rate | 0.0003 | 0.0003 | 0.0003 | 0.0001 | 0.0003 | 0.0003 | 0.0003 |
| Actor number blocks | 256 | 256 | 256 | 128 | 256 | 256 | 512 |
| Actor number of blocks | 4 | 2 | 2 | 1 | 1 | 1 | 3 |
| Policy update delay | 3 | 3 | 1 | 1 | 1 | 1 | 1 |
| Initial temperature | 0.01 | 0.01 | 1 | 0.01 | 1.0 | 0.1 | - |
| Temperature learning rate | 0.0003 | 0.0003 | 0.0003 | 0.0001 | 0.0003 | 0.0003 | - |
| Target entropy | $|A|/2$ | $|A|/2$ | $|A|/2$ | $|A|/2$ | $|A|/2$ | $|A|/2$ | - |
| Target network momentum | 0.005 | 0.005 | 0.005 | 0.005 | 0.005 | 0.005 | 1.0 |
| Target network update frequency | 1 | 1 | 1 | 1 | 1 | 1 | 250 |
| Number of critics | 2 | 2 | 2 | 2 on MuJoCo / HB 1 on DMC / Myo | 2 | 2 | 2 |
| Discount | Heuristic | Heuristic | Heuristic | Heuristic | 0.99 | 0.99 | 0.99 |
| Optimizer | Adam | AdamW | Adam | Adam | AdamW | AdamW | AdamW |
| Weight Decay | - | 0.01 | - | - | 0.0001 | 0.0001 | 0.0001 |
| Categorical Support | [-5, 5] | - | - | [-5, 5] | - | [-10, 10] | - |
| Critic loss | CE | MSE | MSE | CE | MSE | CE | L1 Loss |
| UTD | 2 | 2 | 1 | 2 | 2 BRO Small 10 BRO | 2 | 1 |
| Batch size | 256 | 256 | 256 | 256 | 128 | 1024 | 256 |
| Action repeat | MuJoCo: 1 DMC: 2 HB: 2 Myo: 2 | MuJoCo: 1 DMC: 2 HB: 2 Myo: 2 | MuJoCo: 1 DMC: 2 HB: 2 Myo: 2 | MuJoCo: 1 DMC: 2 HB: 2 Myo: 2 | MuJoCo: 1 DMC: 1 HB: 1 Myo: 1 | MuJoCo: 1 DMC: 2 HB: 2 Myo: 2 | MuJoCo: 1 DMC: 2 HB: 2 Myo: 2 |

Table 2: Hyperparameters for vision-based RL tasks.

| Hyperparameter | XQC | DRQ-V2 |
|---|---|---|
| Block design | Dense(dim)
BN
ReLU | Dense(dim)
ReLU |
| Critic learning rate | 0.0001 | 0.0001 |
| Critic hidden dim | 1024 | 1024 |
| Critic number of blocks | 4 | 2 |
| Actor learning rate | 0.0001 | 0.0001 |
| Actor hidden dim | 1024 | 1024 |
| Actor number of blocks | 4 | 3 |
| Target network momentum | 0.01 | 0.01 |
| Target network update frequency | 1 | 1 |
| Number of critics | 2 | 2 |
| Discount | 0.99 | 0.99 |
| Optimizer | Adam | Adam |
| Categorical Support | [-5, 5] | - |
| Critic Loss | CE | MSE |
| UTD | 0.5 | 0.5 |
| Batch size | 256 | 256 |
| Replay buffer capacity | 1M | 1M |
| N-step returns | 3 | 3 |
| Feature dim | 50 | 50 |
| Exploration stddev. clip | 0.3 | 0.3 |
| Action Repeat | 2 | 2 |
| Exploration stddev. schedule (Defined by task difficulty) | easy: linear(1.0, 0.1, 100K)
medium: linear(1.0, 0.1, 500K)
hard: linear(1.0, 0.1, 2M) | easy: linear(1.0, 0.1, 100K)
medium: linear(1.0, 0.1, 500K)
hard: linear(1.0, 0.1, 2M) |

## B    ENVIRONMENT AGGREGATION DETAILS

As the magnitude of returns varies across environments, we normalize them for comparability before aggregating (Agarwal et al., 2021). We normalize scores to be between 0 and 1. Where the normalization protocols are benchmark-specific and follow standard practice.

**For MuJoCo and Humanoidbench,** we compute the normalized score as

$$\hat{x} = \frac{x - \text{Random Score}}{\text{Target Score} - \text{Random Score}},$$

where the random scores are obtained using a uniformly random policy (Fu et al., 2020). Target scores are taken from a trained TD3 policy in MuJoCo, and are provided by the authors for HB, where they represent the threshold required to mark a task as solved.

**For DMC tasks,** we normalize by dividing the final score by 1000, the maximum achievable return.

**MyoSuite** tasks require no normalization, as performance is already expressed in percentage-based success rates.

## C    BASELINES

In this section, we briefly describe how the results were collected for every baseline we present in this work. Additionally, all hyperparameters are listed in Appendix A.

**SIMBA-V2 (Lee et al., 2025b).**    We used the results made publicly available on the official GitHub repository. The results are based on 10 seeds. For SIMBA-V2 (small) we ran the code from the official codebase ourselves, for 10 seeds.

**CROSSQ+WN (Palenicek et al., 2025).**    We ran all experiments ourselves using our codebase for 10 seeds.

**BRO (Nauman et al., 2024).**    We used the publicly available results on the official SIMBA-V2 GitHub repository. We only considered the 'small' version of BRO, which uses a UTD ratio of 2. The results are based on 5 seeds.

**MRQ (Fujimoto et al., 2025).**    For MuJoCo and DMC, we used the results provided by SIMBA-V2 on their official GitHub repository, which are based on 10 seeds. We conducted experiments for Myo and HB ourselves, by running the official MRQ codebase using 5 random seeds due to computational reasons. We matched the action repeat used in our experiments to ensure a fair comparison.

**SAC (Haarnoja et al., 2018).**    We ran all experiments ourselves using our codebase. We used the default SAC hyperparameters and ran 5 seeds for every environment.

**BRC (Nauman et al., 2025).**    Using the official BRC GitHub codebase, we experiment ourselves for 3 seeds, due to computational constraints. While we used their default settings as reported in the paper, reducing the parameters to 64M, since Nauman et al. (2025) noted that using more than 64M parameters provides no benefit for single-task settings. Additionally, we used the same action repeat environment wrapper used in all our other experiments, ensuring a fair comparison.

**DRQ-V2 (Hiraoka et al., 2021).**    We used the results reported in the offical GitHub repository based on 10 seeds. Our vision-based XQC experiments are based on the DRQ-V2 codebased for a fair comparison.

# D ALL TRAINING CURVES: REINFORCEMENT LEARNING FROM VISION-BASED DMC ENVIRONMENTS

Results from RL on the vision-based DMC benchmarks. We compare to DRQ-V2 (Yarats et al., 2022).

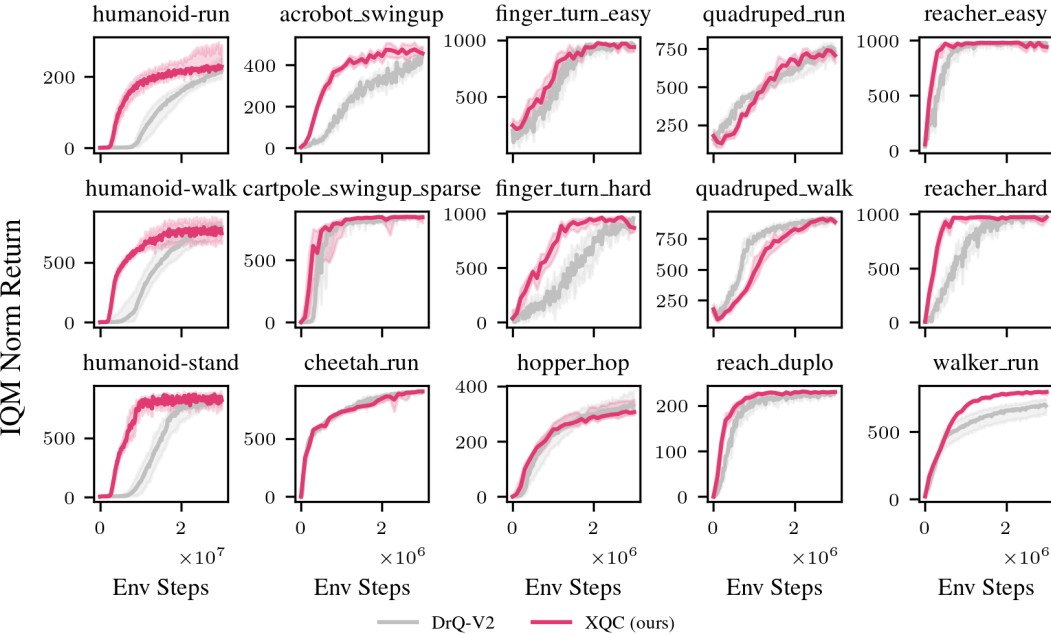

Figure 11: XQC and DRQ-V2 training curves for each of the 15 vision-based DMC tasks. We show the IQM and 90% SBCI aggregated over 10 seeds per environment.

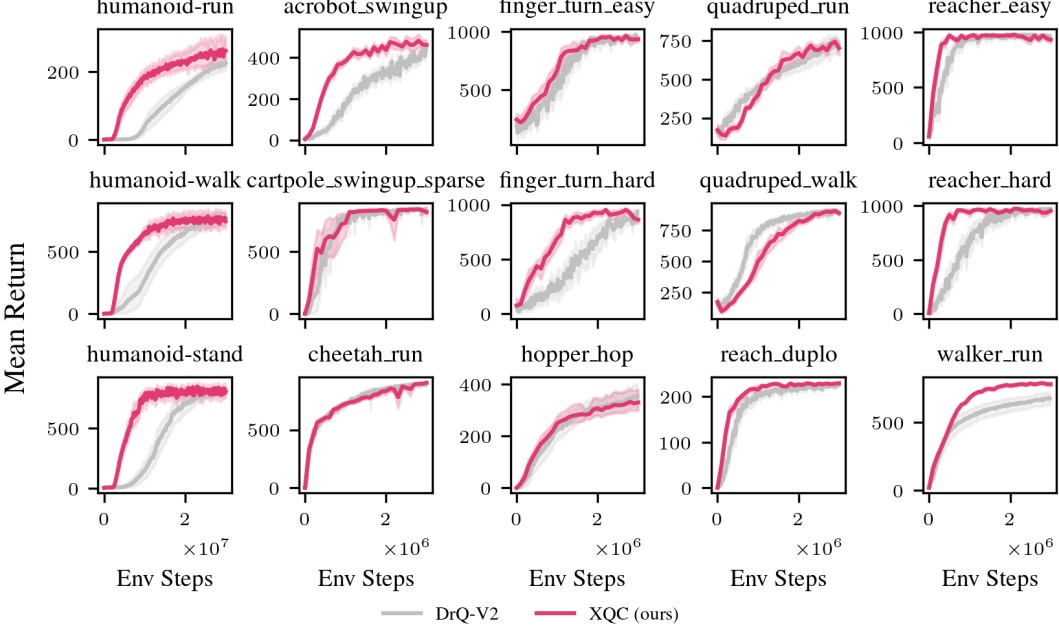

Figure 12: XQC and DRQ-V2 training curves for each of the 15 vision-based DMC tasks. We show the mean and 90% SBCI aggregated over 10 seeds per environment.

# E ALL TRAINING CURVES: REINFORCEMENT LEARNING FROM PROPRIOCEPTION

All results for the proprioception continuous control benchmarking tasks with IQM and mean.

## E.1 HUMANOIDBENCH

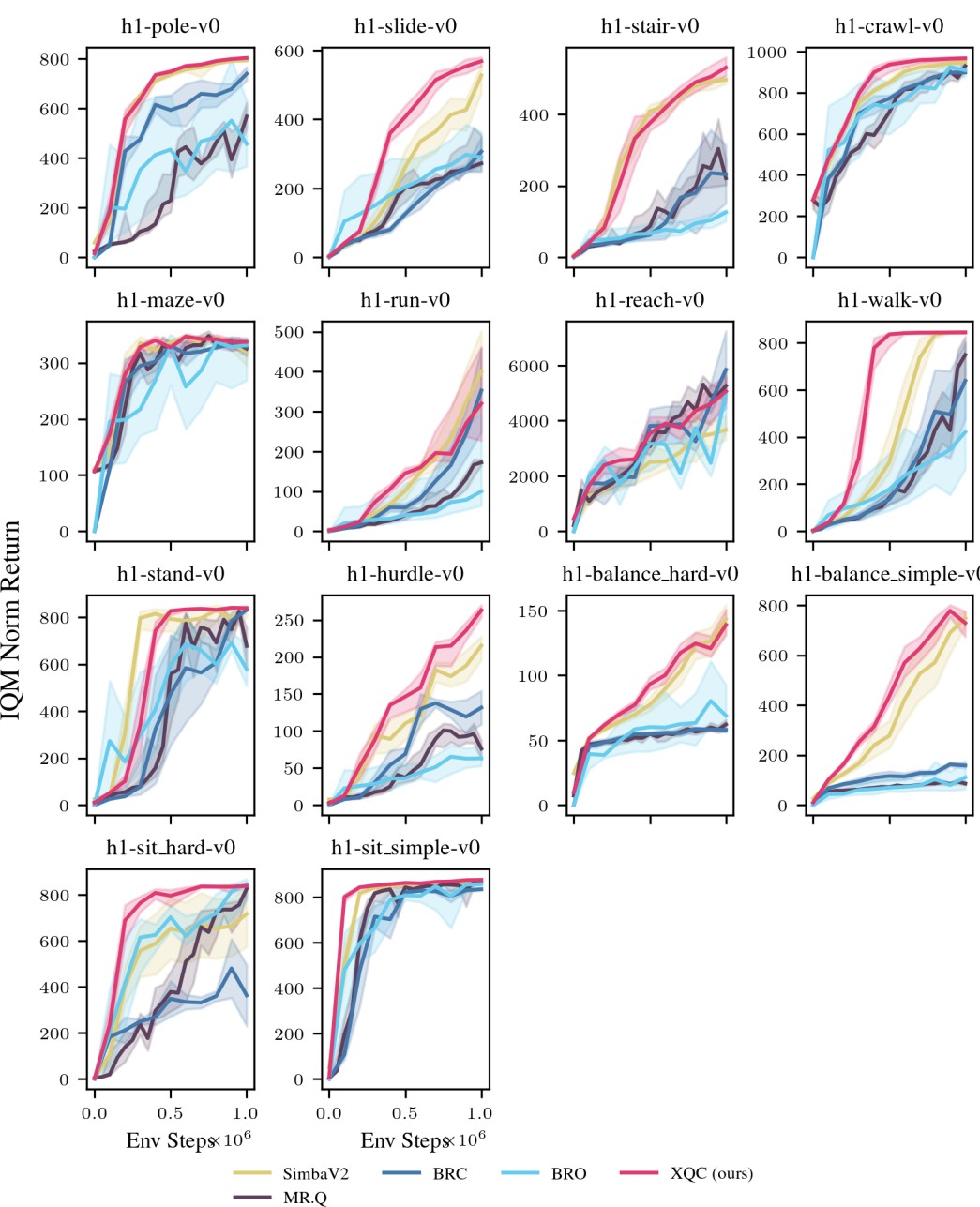

Figure 13: XQC and baseline training curves for each of the 14 HB tasks. We show the IQM and 90% SBCI aggregated per environment.

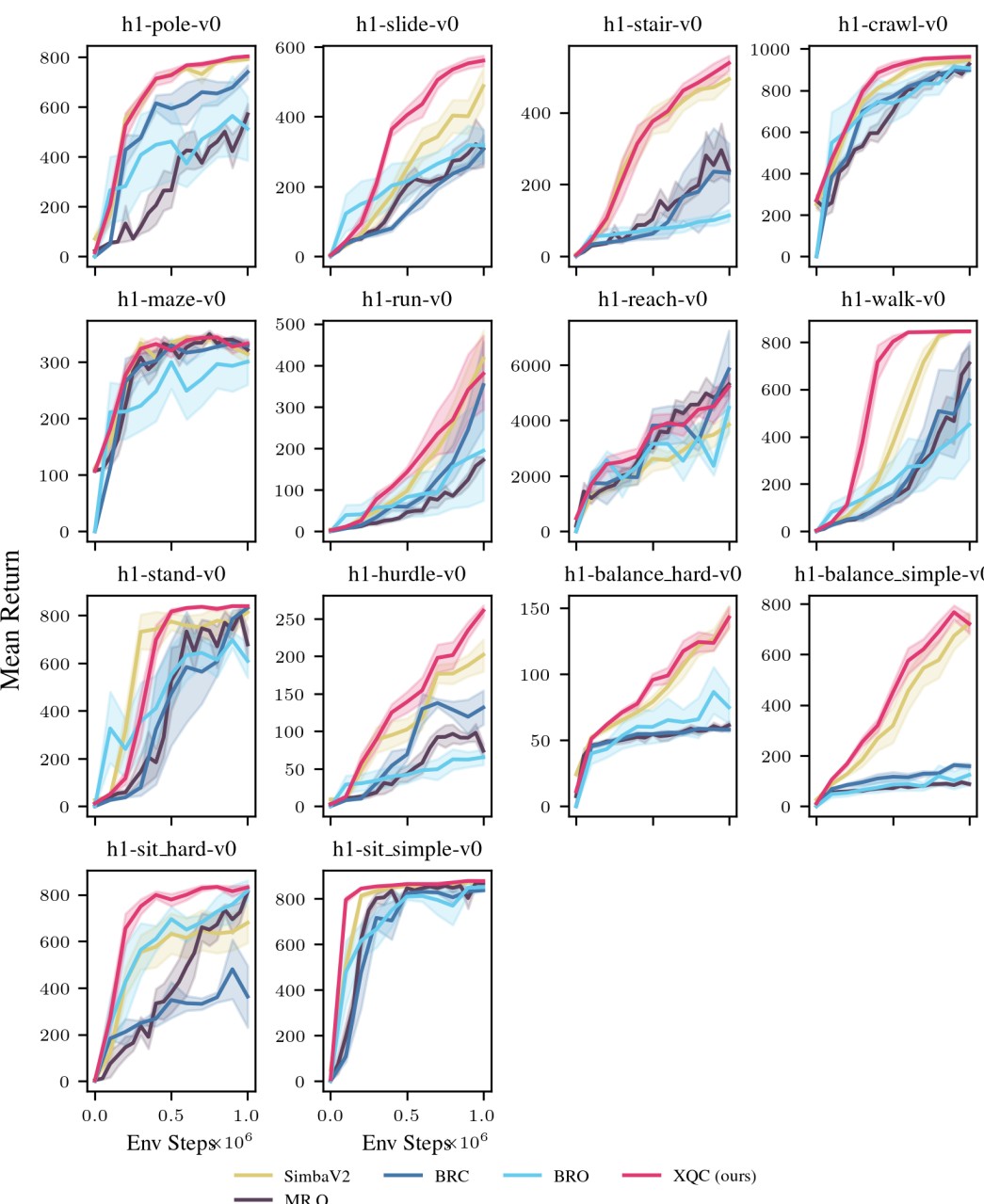

Figure 14: XQC and baseline training curves for each of the 14 HB tasks. We show the mean and 90% SBCI aggregated per environment.

## E.2 DEEPMIND CONTROL SUITE

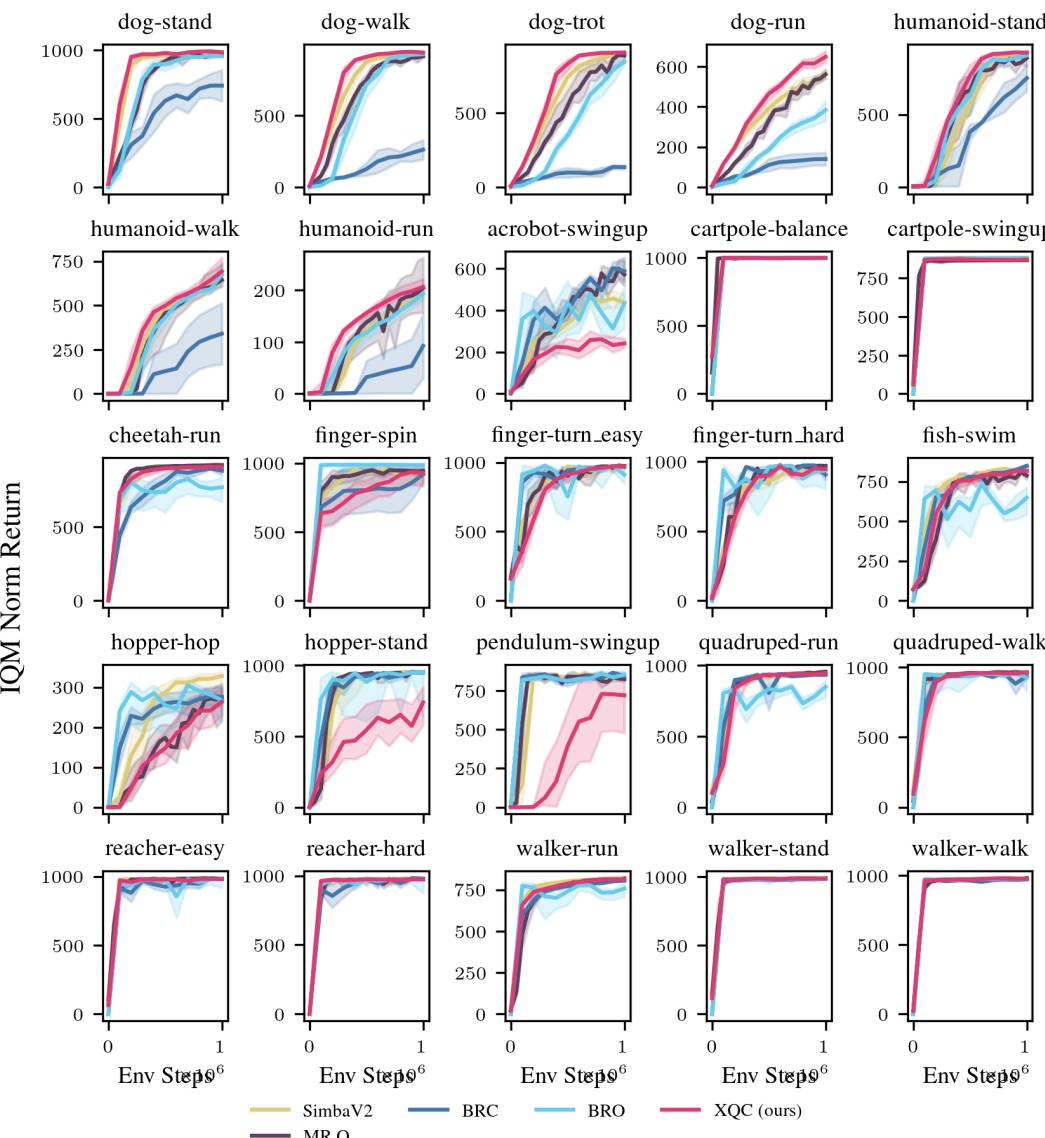

Figure 15: XQC and baseline training curves for each of the 25 DMC tasks. We show the IQM and 90% SBCI aggregated per environment.

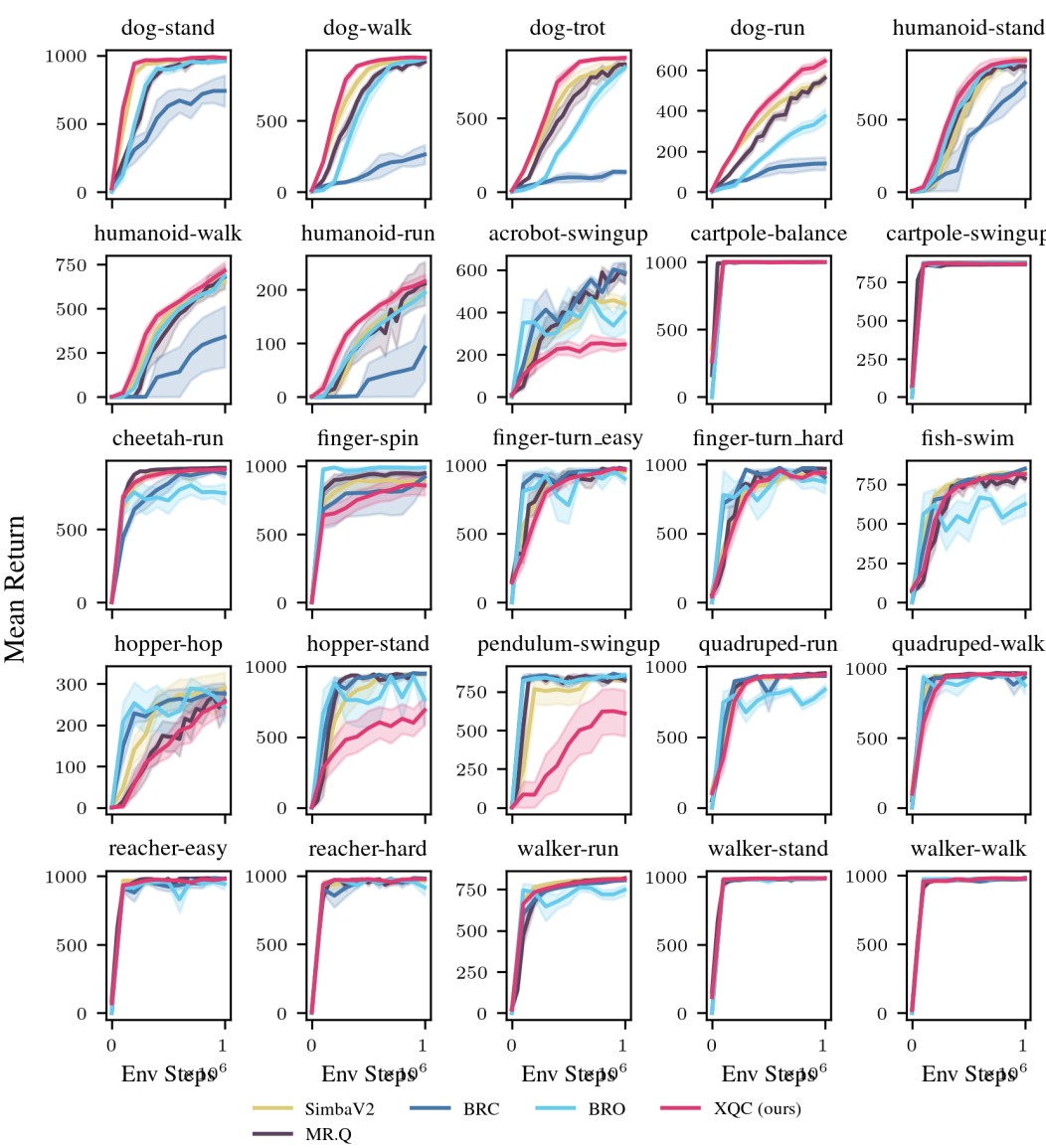

Figure 16: XQC and baseline training curves for each of the 25 DMC tasks. We show the mean and 90% SBCI aggregated per environment.

### E.3 MYOSUITE

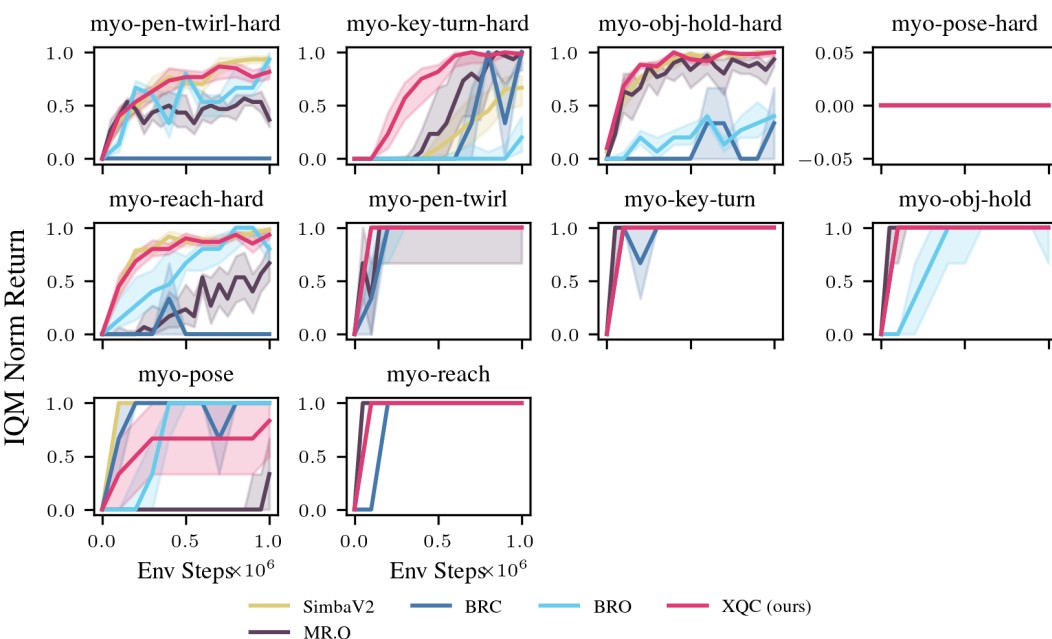

Figure 17: XQC and baseline training curves for each of the 10 Myo tasks. We show the IQM and 90% SBCI aggregated per environment.

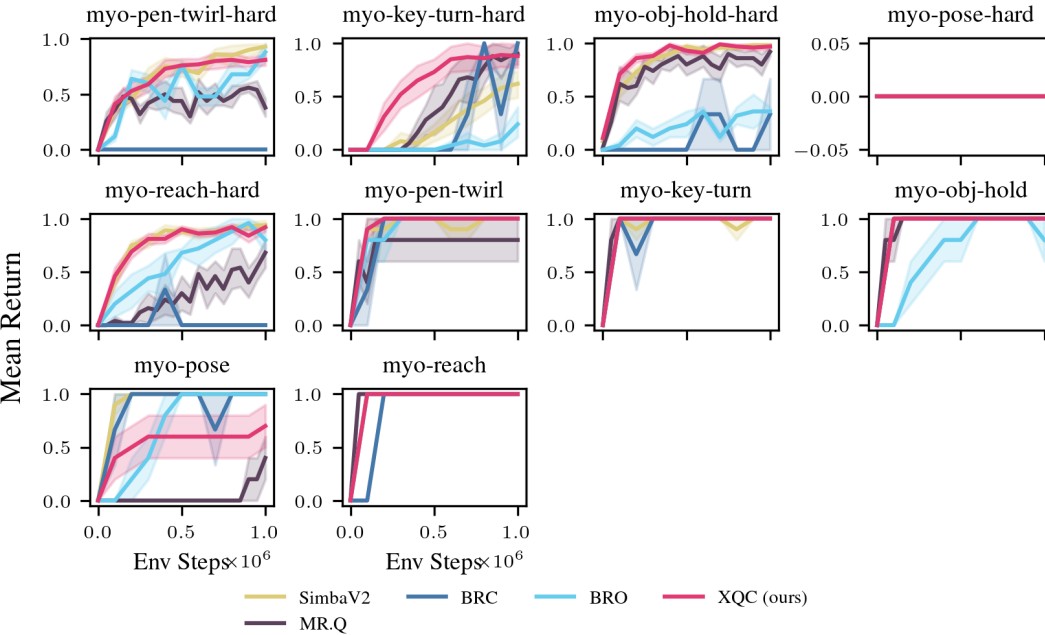

Figure 18: XQC and baseline training curves for each of the 10 Myo tasks. We show the mean and 90% SBCI aggregated per environment.

### E.4 MuJoCo Benchmark

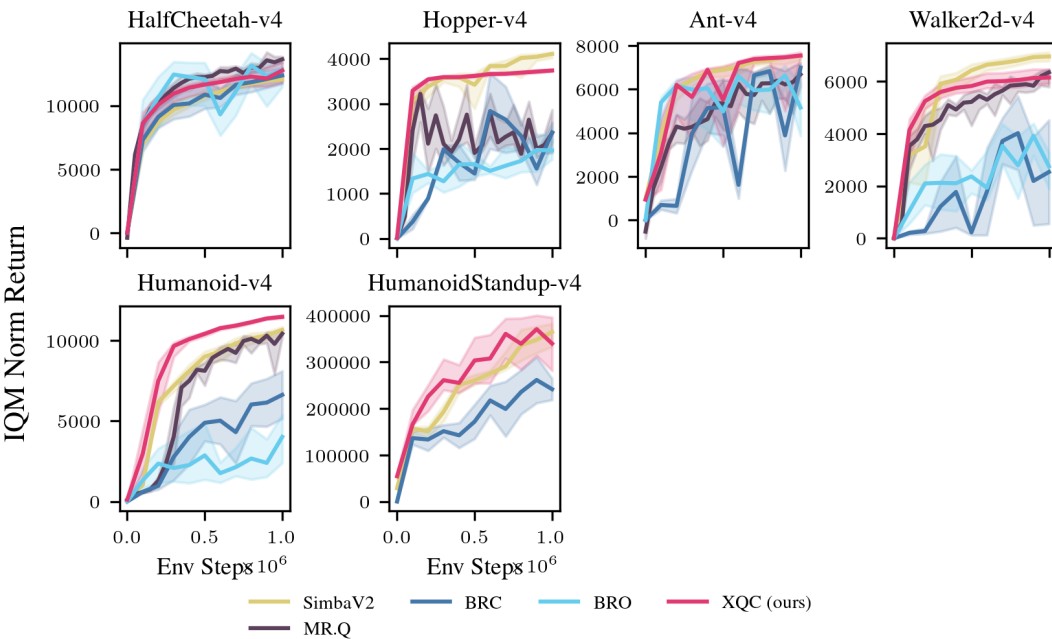

Figure 19: XQC and baseline training curves for each of the 6 MuJoCo tasks. We show the IQM and 90% SBCI aggregated per environment.

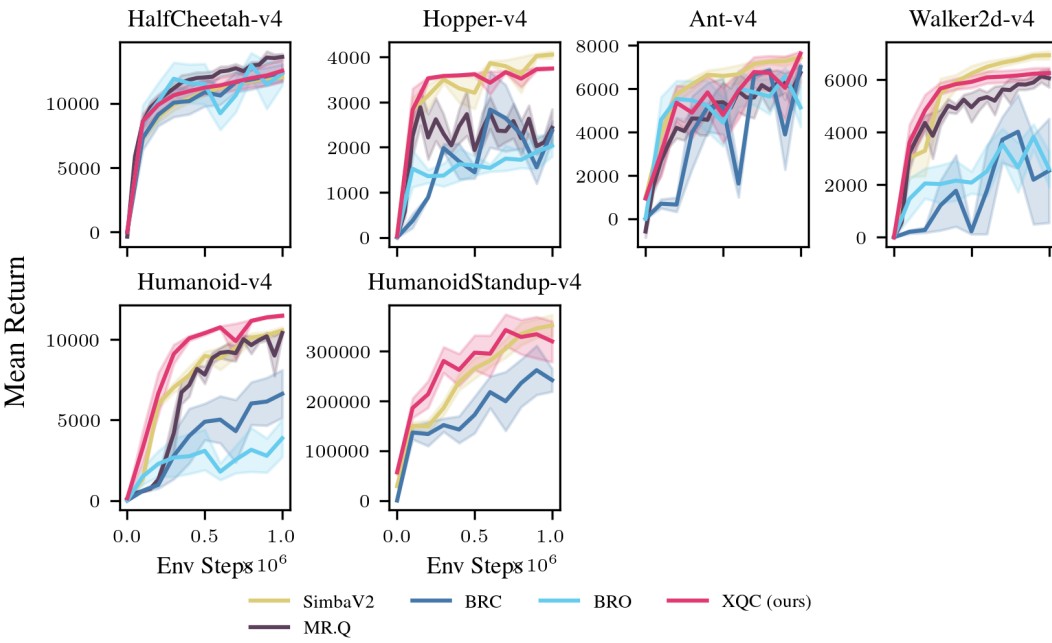

Figure 20: XQC and baseline training curves for each of the 6 MuJoCo tasks. We show the mean and 90% SBCI aggregated per environment.

# F  PLASTICITY METRICS

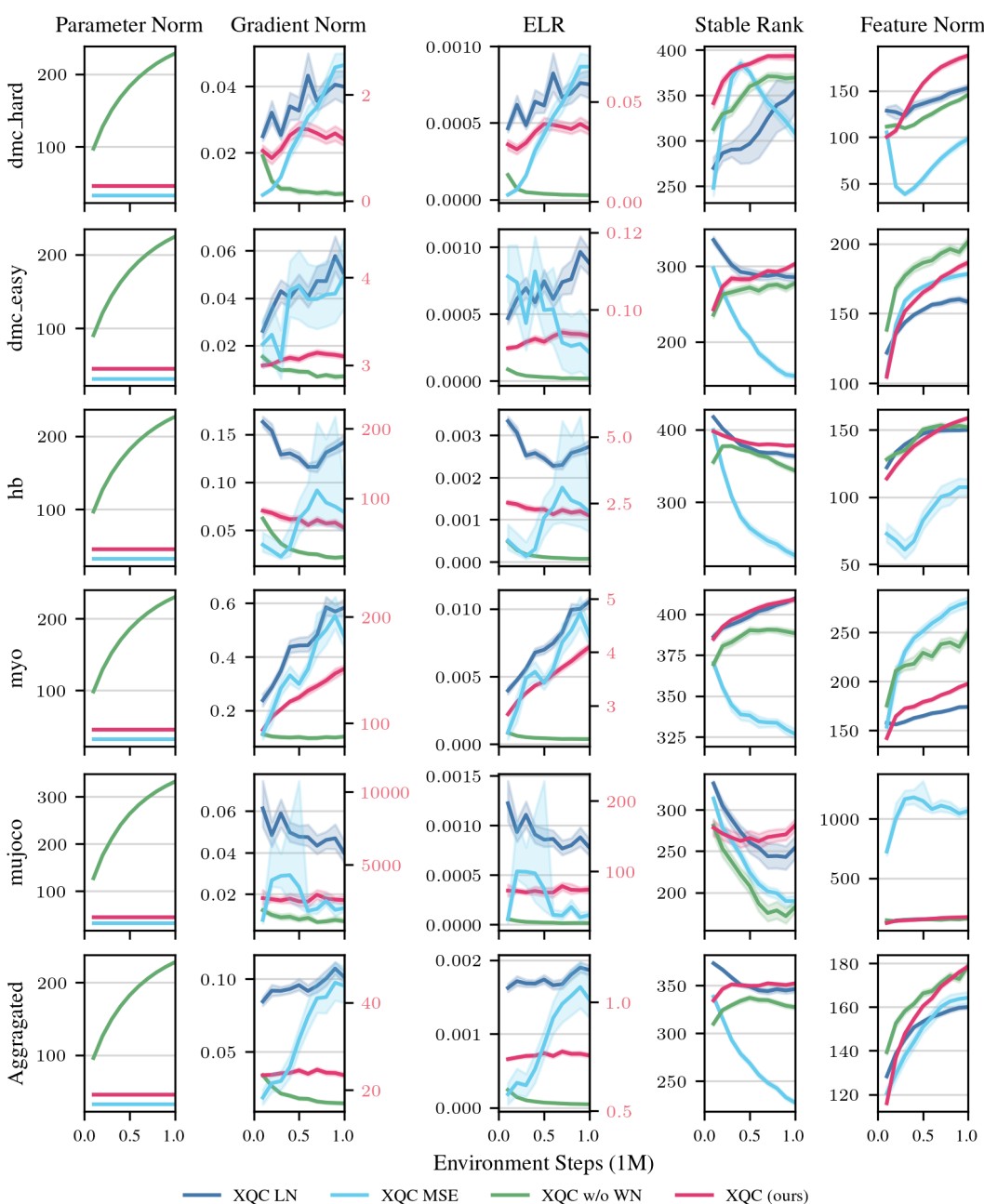

Figure 21: Per benchmark plasticity metrics for XQC and architectural ablations.

# G ARCHITECTURE ABLATIONS

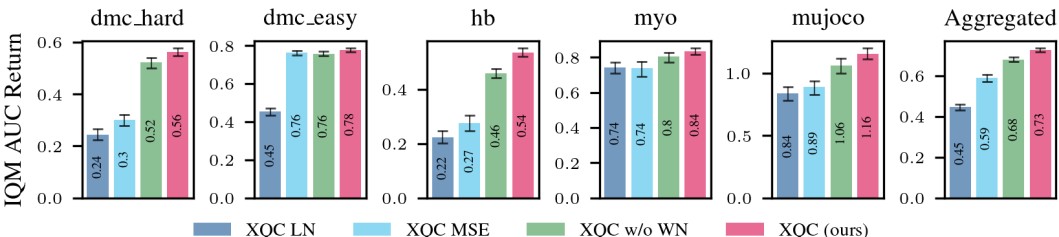

Figure 22: **Ablation study confirms the necessity of all three of XQC's components.** We compare the full XQC algorithm against three variants: one replacing BN with LN (XQC LN), one replacing the CE loss with an MSE loss (XQC MSE), and one without WN (XQC w/o WN). Each component's removal results in a significant performance drop, demonstrating their synergistic contribution.

# H XQC UTD SCALING TRAINING CURVES

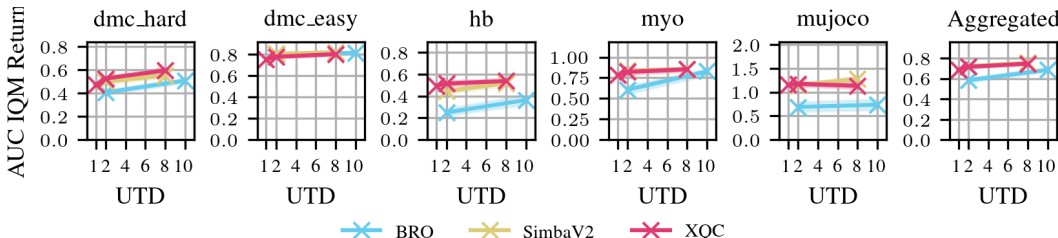

Figure 23: **XQC stably improves with increased UTD ratios.** We compare IQM AUC for XQC trained with UTD ratios $\in \{1, 2, 8, 16\}$. Performance consistently improves with more updates, showcasing the stability of the learning process.

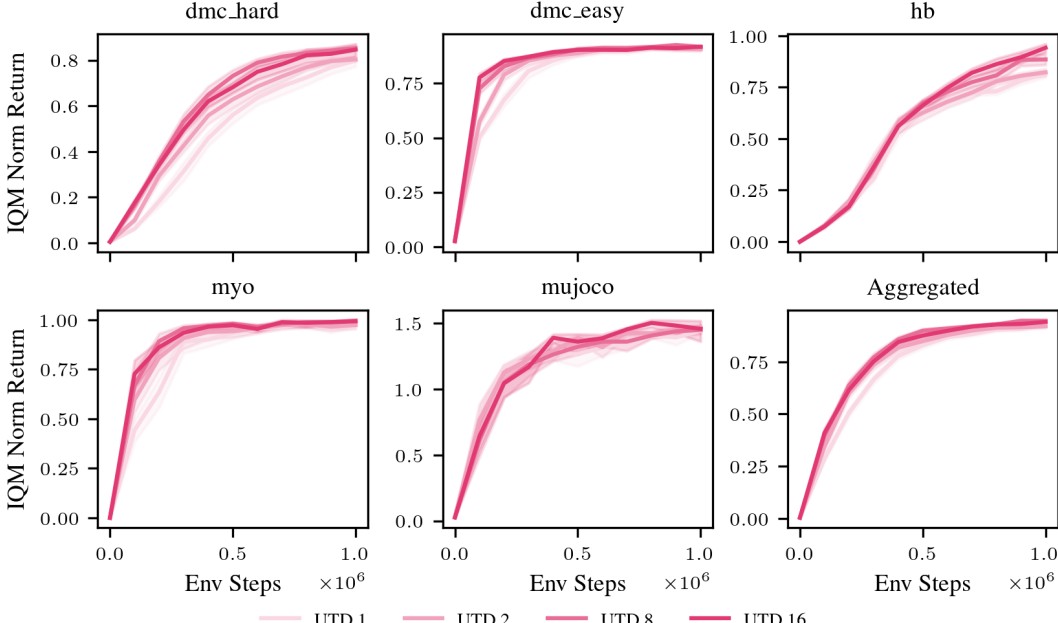

Figure 24: UTD scaling. We present the area under the curve of the IQM Norm Return. This measure captures fast and stable learning simultaneously.

# I  ADDITIONAL CONDITIONING RESULTS

Here we present additional conditioning results for alternative architectural components requested by the reviewers during the rebuttal. We ablate combining XQC with skip connections (with network depth in $\in \{4, 6\}$, different batch sizes $\in \{128, 256, 512\}$ and weight decay $1e - 4$. Further, we investigate how depth scaling influences conditioning for dense MLPs with number of layers $\in \{2, 4, 6\}$. The chosen architectural ablations on top of XQC do not meaningfully influence conditioning or performance. The overall finding is in line with our main finding, that good conditioning correlates with good RL performance.

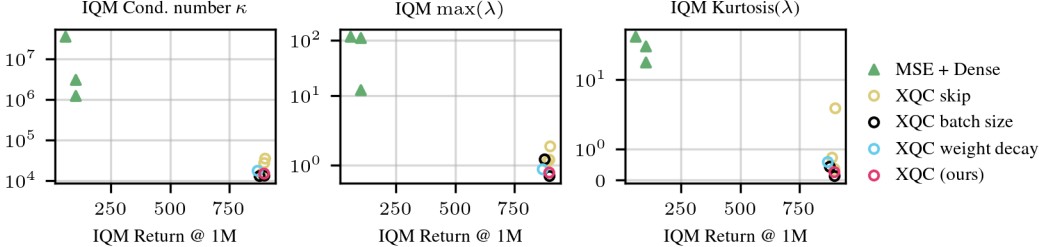

Figure 25: The condition numbers and maximum eigenvalues against the return at 1M steps on DMC `dog-trot`. Architectures with lower condition numbers and lower maximum eigenvalues tend to have better final returns.

## J    THEORETICAL ANALYSIS

The section details the proofs for bounding the gradient norms and Hessian condition numbers.

**Lemma 1.** *For the loss $\mathcal{L}(\boldsymbol{\theta}, \mathcal{D}) = l(\boldsymbol{Y}, \boldsymbol{f_\theta}(\boldsymbol{X}))$, if $\boldsymbol{f}$ is $L_{\boldsymbol{f}}$ Lipschitz in the L2 norm with respect to $\boldsymbol{\theta}$, the L2 norm of the gradient has the following upper bound,*

$$||\nabla_{\boldsymbol{\theta}} \mathcal{L}(\boldsymbol{\theta}, \boldsymbol{y}, \boldsymbol{x})||_2 \leq L_{\boldsymbol{f}} \cdot ||\nabla_{\boldsymbol{f}} l(\boldsymbol{y}, \boldsymbol{f_\theta}(\boldsymbol{x}))||_2. \tag{6}$$

*Proof.* Using the chain rule and the Cauchy-Schwarz inequality,

$$||\nabla_{\boldsymbol{\theta}} \mathcal{L}(\boldsymbol{\theta}, \boldsymbol{y}, \boldsymbol{x})||_2 \leq ||\nabla_{\boldsymbol{f}} l(\boldsymbol{y}, \boldsymbol{f_\theta}(\boldsymbol{x}))||_2 \cdot ||\nabla_{\boldsymbol{\theta}} \boldsymbol{f_\theta}(\boldsymbol{x})||_2 \leq ||\nabla_{\boldsymbol{f}} l(\boldsymbol{y}, \boldsymbol{f_\theta}(\boldsymbol{x}))||_2 \cdot L_{\boldsymbol{f}}. \tag{7}$$

$\square$

*Proof of Proposition 1.* Standard calculus. $\square$

*Proof of Proposition 2.* Standard calculus, and then using the difference of two categorical probability vectors (on a simplex) to bound the largest squared error as 2. $\square$

*Proof of Theorem 1.* Combine Lemma 1, Proposition 2, and the constrained parameter norm for the definition of the gradient update in Definition 1 to obtain an upper bound. $\square$

**Lemma 2.** *For a symmetric matrix $\boldsymbol{A} \in \mathbb{R}^{m \times m}$ with ranked eigenvalues $\lambda_1^A \leq \cdots \leq \lambda_m^A$, then the eigenvalues of the sum of two such matrices $\boldsymbol{C} = \boldsymbol{A} + \boldsymbol{B}$, then $\lambda_1^A + \lambda_1^B \leq \lambda_1^C$ and $\lambda_m^A + \lambda_m^B \geq \lambda_m^C$. This result holds for all finite sums.*

*Proof.* Weyl's theorem applied to the sum of two Hermitian matrices (Bodmann, 2012). $\square$

**Proposition 5.** *The mean squared error loss, $l(\boldsymbol{y}, \hat{\boldsymbol{y}}) = \frac{1}{2}||\boldsymbol{y} - \hat{\boldsymbol{y}}||_2^2$, $\boldsymbol{y} \in \mathbb{R}^d$ has a constant Hessian and therefore constant Hessian eigenvalues $\lambda$,*

$$\nabla_{\hat{\boldsymbol{y}}}^2 l(\boldsymbol{y}, \hat{\boldsymbol{y}}) = \boldsymbol{I}_d, \quad \lambda_{1:d} = 1. \tag{8}$$

*Proof.* Standard calculus. $\square$

**Proposition 6.** *The cross entropy loss, $l(\boldsymbol{y}, \hat{\boldsymbol{y}}) = -\sum_{i=1}^d y_d \log \hat{y}_d$ has the following Hessian and eigenvalue bounds given the model $\hat{\boldsymbol{y}} = Softmax(\boldsymbol{f_\theta}(\boldsymbol{x}))$ where $y_i \geq \epsilon$,*

$$\nabla_{\boldsymbol{f}}^2 l(\boldsymbol{y}, \boldsymbol{f_\theta}(\boldsymbol{y})) = \mathrm{diag}(\hat{\boldsymbol{y}}) - \hat{\boldsymbol{y}}\hat{\boldsymbol{y}}^\top, \quad 0 \leq \lambda_i \leq 1, \tag{9}$$

*as $\sum_{i=1}^d y_i = 1, 0 \leq y_i \leq 1$. The Hessian is singular due to the loss of degree-of-freedom in categorical probabilities.*

*Proof.* Standard calculus. $\square$

For Proposition 3 and 4, we use that the objective's Hessian can be decomposed using the chain rule,

$$\nabla_{\boldsymbol{\theta}}^2 \mathcal{L}(\boldsymbol{\theta}, \boldsymbol{y}, \boldsymbol{x}) = \nabla_{\boldsymbol{\theta}} \boldsymbol{f_\theta}(\boldsymbol{x})^\top \nabla_{\boldsymbol{f}}^2 l(\boldsymbol{y}, \boldsymbol{f_\theta}(\boldsymbol{x})) \nabla_{\boldsymbol{\theta}} \boldsymbol{f_\theta}(\boldsymbol{x}) + \nabla_{\boldsymbol{f}} l(\boldsymbol{y}, \boldsymbol{f_\theta}(\boldsymbol{x}) \nabla_{\boldsymbol{\theta}}^2 \boldsymbol{f_\theta}(\boldsymbol{x}),$$
$$= \boldsymbol{g_\theta}(\boldsymbol{x})^\top \boldsymbol{H}_l(\boldsymbol{\theta}, \boldsymbol{x}, \boldsymbol{y}) \boldsymbol{g_\theta}(\boldsymbol{x}) + \boldsymbol{g}_l(\boldsymbol{\theta}, \boldsymbol{x}, \boldsymbol{y})^\top \boldsymbol{H_\theta}(\boldsymbol{x}). \tag{10}$$

*Proof for Proposition 3.* The first term of Equation 10 has a rank of 1 as it's an outer product, and $\boldsymbol{g}(\boldsymbol{\theta}, \boldsymbol{x})^\top \boldsymbol{g}(\boldsymbol{\theta}, \boldsymbol{x}) = ||\boldsymbol{g}||_2^2 \leq L_{\boldsymbol{f}}^2$, so its eigenvalues $\lambda_i \in [0, L_{\boldsymbol{f}}^2]$. Using Assumption 1, the eigenvalues of the second term are bounded by $[-m|g|_{\max}\lambda_m^f, m|g|_{\max}\lambda_m^f]$. As the gradient elements cannot be upper bounded (i.e., Proposition 1), the Hessian of the loss has eigenvalue range $[\mu^2 - 2m|g|_{\max}\lambda_m^f, \mu^2 + 2m|g|_{\max}\lambda_m^f]$, which leads to an unbounded condition number due to both the largest eigenvalue $\to \infty$ and the the case that the smallest eigenvalue is 0 when adding eigenvalues from both terms due to Weyl's theorem (Lemma 2). $\square$

*Proof for Proposition 4.* The first term in Equation 10 has eigenvalue bounds $[0, L_{\bm{f}}^2]$ (see previous proof). It's positive semi-definite so we know 0 is a lower bound on the eigenvaluse. Since the max eigenvalue is non-zero, we know the Frobenius norm of $\bm{g}$ is greater or equal to than the trace of the outer product, and the trace is also the sum of eigenvalues, so we can bound the (largest) non-zero eigenvalue by $L_{\bm{f}}^2$. The second term in Equation 10 has range $\lambda_i \in [-2\lambda_m^f, 2\lambda_m^f]$, as they are bounded by $[-\sum_i |g_i| \lambda_m^H, \sum_i |g_i| \lambda_m^H]$ and $0 \geq |g_i| \geq 1$, $\sum_i g_i = 0$. With Weyl's theorem (Lemma 2) we have $\lambda \in [\mu^2 - 2\lambda_m^H, \mu^2 + 2\lambda_m^H + L_{\bm{f}}^2]$. If $\mu^2 = 2\lambda_m^H + \epsilon$, then we have $\lambda \in [\epsilon, 4\lambda_m^H + \epsilon + L_{\bm{f}}^2]$, so

$$\kappa \leq \frac{4\lambda_m^f + L_{\bm{f}}^2 + \epsilon}{\epsilon}$$

which concludes the proof. Unsurprisingly, the upper bound is only finite if the regularization ensures positive definiteness of the objective's Hessian. $\square$

