# OpenReview forum: "XQC: Well-conditioned Optimization Accelerates Deep Reinforcement Learning"
_ICLR.cc/2026/Conference — ICLR 2026 Poster_

### Official Review · Reviewer_79k1 · 2025-10-28

**Soundness:** 3
**Presentation:** 4
**Contribution:** 3
**Rating:** 8
**Confidence:** 4

**Summary:**

The paper starts with a systematic evaluation of three design choices in RL (normalization layer, MSE-CE loss, and weight normalization), based on the smoothness of the critic loss curvature. Their results suggest that the combination of batch normalization, CE loss, and weight normalization yields the smoothest loss curvature and thus stable optimization process. Theoretical insights are additionally provided on why CE loss generates much more stable gradients than standard MSE loss.

Based on the results, authors propose a simple architecture design that encorporates the three components. With significantly smaller number of parameters compared to prior works, their architecture shows competitive or even superior performance in diverse benchmarks comprising up to 55 tasks. Their results also extend to vision-based RL, where the architecture leads to significant improvement of DrQ-v2, especially in challenging humanoid environments.

Finally, their ablation results reveal that switching on of the components lead to degradation of performance, validating their design choices.

**Strengths:**

- **Well-motivated design choices**. Authors have verified their design choices based on thorought and systematic evaluation of the smoothness of the critic loss curvature.
- **Strong performance across diverse benchmarks**. The proposed architecture shows strong results compared to prior SOTAs in total 55 tasks.

**Weaknesses:**

- I only see minor questions and weaknesses; please see the questions.

**Questions:**

- I believe comparing the actual train/inference speed (FPS) with prior SOTAs would also be meaningful, as it directly reflect the speed expected in practice, and may differ from FLOPs depending on aspects such as number of layers.
- To my understanding, IQM was used to remove outlier tasks in Atari benchmark [1], and it feels unnatural to use IQM for learning curves of individual tasks. I don't necessarily go fully against using IQM, but I suggest the authors provide average results for both individual learning curves and aggregate results along with IQM.
- In Figure 8, XQC LN seems to have twice as much ELR as XQC. Would it be possible that this difference caused the performance drop shown in Figure 9 (right)? Have you tried decreasing the base learning rate to $1.5e-4$, half of default value of XQC?

---

> ### Author Response · Authors · 2025-11-22
> **Response to Reviewer 79k1**
>
> We thank the reviewer for their very positive and constructive review. We were very pleased about the excellent presentation score as well as the acknowledgment of our well-motivated design choices and strong performance.
>
> In the following we want to answer each of the reviewer’s questions in detail.
>
> > I believe comparing the actual train/inference speed (FPS) with prior SOTAs would also be meaningful, as it directly reflect the speed expected in practice, and may differ from FLOPs depending on aspects such as number of layers.
>
> We have added two columns to Figure 9, with training FPS and evaluation FPS of the MuJoCo Humanoid measured on an RTX 4090 workstation. XQC has 60% higher training FPS than the next quickest baseline. In terms of evaluation FPS it is slightly slower than SimbaV2. However, we want to emphasize that in practice the difference in evaluation speed is negligible, since all baselines are well above 1000 FPS, which is enough for even high frequency control tasks.
> Originally, we focused on reporting network parameters and FLOPs/s, because these metrics are more objective, as they are hardware-independent. However, we agree that FPS is a valuable addition to the efficiency analysis, especially for practitioners.
>
> ---
>
> > To my understanding, IQM was used to remove outlier tasks in Atari benchmark [1], and it feels unnatural to use IQM for learning curves of individual tasks. I don't necessarily go fully against using IQM, but I suggest the authors provide average results for both individual learning curves and aggregate results along with IQM.
>
> Agarwal et. al’s motivation to propose IQM as a metric for RL was that “[in contrast to the mean, IQM is] robust to outlier scores but more statistically efficient than median” and this is the reason why we opted for it.
> However, we have added individual average training curves to Appendix D & E as requested. The mean curves show the same qualitative trends and results. Since the main paper is already rather dense with many information dense plots, we believe additionally including average results in the main part would make the paper cluttered and less readable, hence, the inclusion in the Appendix for interested readers. We are happy to discuss this further, should the reviewer disagree.
>
> ---
>
> > In Figure 8, XQC LN seems to have twice as much ELR as XQC. Would it be possible that this difference caused the performance drop shown in Figure 9 (right)? Have you tried decreasing the base learning rate to , half of default value of XQC?
>
> We have added the requested ablation to Figure 10. We find that for LN with half the learning rate, performance degrades further. This rules out the mismatched ELR hypothesis and backs our loss landscape conditioning analysis (Figure 4): LN induces a significantly worse-conditioned landscape that cannot be fixed by simply tuning the learning rate.

---

### Official Review · Reviewer_ATvv · 2025-10-29

**Soundness:** 3
**Presentation:** 3
**Contribution:** 3
**Rating:** 6
**Confidence:** 4

**Summary:**

This paper introduces XQC, a simple actor-critic architecture that improves sample efficiency in deep reinforcement learning through well-conditioned optimization. Rather than relying on model scaling or complex architectures, XQC achieves stability and efficiency by combining batch normalization, weight normalization, and a categorical cross-entropy loss, which together yield lower Hessian condition numbers and bounded gradient norms.

Theoretical analysis and large-scale empirical results - spanning 70 continuous control tasks from proprioceptive and vision-based benchmarks - demonstrate that this synergy leads to superior training stability, plasticity, and state-of-the-art performance while using significantly fewer parameters and compute than prior methods such as BRO or SIMBA-V2.

**Strengths:**

- Studying architectures that allow for scaling actor-critic is timely and important
- The proposed method is simple and seems to offer a more computationally efficient alternative to scaling via BRO or SimBa
- I appreciate the analysis comparing different normalization strategies

**Weaknesses:**

- Limited novelty. The work combines puzzle pieces from previous works (batch norm + weight norm was used in [1], weight norm + categorical loss was used in [2], layer norm + categorical loss was used in [3]). I still appreciate showcasing that this particular combination seems to be most effective for scaling actor-critic.

**Questions:**

1. Why do the authors use an action repeat wrapper in the humanoid bench? The humanoid bench uses joint position control, which does not make much sense with action repeat. Because of how joint position commands work, repeating that same position command multiple times has no new effect (ie. the robot just keeps tracking the same target position, no?)
2.  [4] found that BRO performance significantly changes depending on the action repeat setting. Did the authors test their method without action repeat on benchmarks other than Mujoco?
3.  Is it expected that BRC underperforms BRO? To the best of my understanding, they differ only by using categorical loss instead of MSE. Considering the presented results on the importance of the categorical loss, shouldn't BRC perform similarly to BRO?
4. Do authors have a hypothesis as to why the performance of their approach decreases when using more layers?
5. BRO and SimBa use residual architectures. Could the authors comment on why they opted for standard MLP here? Do authors think that it might matter for more complex environments?
6. It would be interesting to see how DrQ+BRO or DrQ+SimBa performs as compared to DrQ+XQC, considering that BRO and SimBa papers did not include vision-based DMC experiments. Do authors think that these methods would also perform worse than XQC in vision experiments?
7. Could the authors summarize the shortcomings of their theoretical analysis? What are the differences between the assumptions and real-world problems?
8. It would be interesting to see how scaling improves conditioning on its own as compared to using bn + wn + categorical loss.
9. Batch normalization is interesting because the batch statistics depend on the sampled batch (and the variance of these statistics depends on the batch size). To this end, it would be interesting to see how robust XQC is to different batch size settings and whether the increased variance improves the optimization landscape.

Nitpicks:
- line 48 update-to-date -> update-to-data
- line 099 bellmann -> bellman
- line 331 propioceptive -> proprioceptive


[1] Palenicek et al. Scaling CrossQ with Weight Normalization
[2] Lee et al. Hyperspherical Normalization for Scalable Deep Reinforcement Learning
[3] Nauman et al. Bigger, Regularized, Categorical
[4] Voelcker et al. MAD-TD: Model-Augmented Data stabilizes High Update Ratio RL

---

> ### Author Response · Authors · 2025-11-22
> **Answer to Reviewer ATvv**
>
> We thank the reviewer for their very positive and constructive review. We were especially happy that they acknowledged the importance and timeliness of our research.
>
> In the following, we want to answer each of the reviewer’s questions in detail.
>
> > Limited novelty. The work combines puzzle pieces from previous works (batch norm + weight norm was used in [1], weight norm + categorical loss was used in [2], layer norm + categorical loss was used in [3]). I still appreciate showcasing that this particular combination seems to be most effective for scaling actor-critic.
>
> We believe that our work goes far beyond combining puzzle pieces from previous works. Instead, we provide a deep analysis of different common deep RL architecture components and motivate our architecture from an optimization-based perspective. To the best of our knowledge, this is the first work to leverage eigenspectrum analysis in investigating the conditioning of loss landscapes induced through different architectural choices in RL. The resulting architecture challenges the current “bigger is better” trend and extends to vision-based RL.
> We also refer the reviewer to our answer to the reviewer `dowu` regarding this point.
>
> ---
>
> > Why do the authors use an action repeat wrapper in the humanoid bench? The humanoid bench uses joint position control, which does not make much sense with action repeat. Because of how joint position commands work, repeating that same position command multiple times has no new effect (ie. the robot just keeps tracking the same target position, no?)
>
> We chose an action repeat of 2 for comparability reasons, as the SimbaV2 baseline also used it. However, in general, it can make sense to use an action repeat even with joint position commands, as the underlying PD controller needs time to converge to the setpoint. Depending on the dynamical system, control frequency, and position control gains, different action repeats can be better.
>
> ---
>
> > [4] found that BRO performance significantly changes depending on the action repeat setting. Did the authors test their method without action repeat on benchmarks other than Mujoco?
>
> We have added an action repeat = 1 ablation to Figure 10. For XQC, there is no significant performance difference. Overall, ee merely adopted the frame skip proposed by the SimbaV2 paper in order to make a fair comparison to the strongest baseline. According to [4], for BRO an action repeat of 1 performs better, in that sense we used the strongest possible BRO baseline.
>
> ---
>
> > Is it expected that BRC underperforms BRO? To the best of my understanding, they differ only by using categorical loss instead of MSE. Considering the presented results on the importance of the categorical loss, shouldn't BRC perform similarly to BRO?
>
> This is indeed a very interesting point and we were surprised by the results as well. After submission, we talked to the BRC authors and they have since found a bug within their reward normalization which could be the cause for the performance difference. We will re-run the fixed BRC baseline, however, since this is very expensive we do not have the results at the current time.
>
> ---
>
> > Do authors have a hypothesis as to why the performance of their approach decreases when using more layers?
>
> While we see a significant performance increase from depth=2 to 4 (0.664 → 0.727), the perceived decrease from depth 4 to 6 is not significant (0.727 → 0.714) with the overlapping confidence intervals. Visually, it is emphasized by the y-axis zoom in (0.5 → 0.8). As such, performance remains constant.
>
> Considering that in this work we focus on simple and smaller architectures, we did not investigate how scaling to very deep networks performs, but we assume that we would run into common gradient propagation issues, probably requiring the introduction of skip connections. We see the extension to very deep architectures as an interesting direction for future work.
>
> ---
>
> > BRO and SimBa use residual architectures. Could the authors comment on why they opted for standard MLP here? Do authors think that it might matter for more complex environments?
>
> The rationale for not including skip connections is that we did not start our investigation from the perspective of the baselines, but instead built the algorithm from our conditioning investigation and only added necessary components. Overall, we are attempting to keep the architecture as simple as possible while maximizing performance.
>
> We have, however, added an ablation with residual connections and different network depths to Figure 10. Further, we added an investigation of the loss-landscape conditioning of skip connections to Appendix I. We find that skip connections do not have a significant effect on XQC’s performance or loss-landscape conditioning.

---

> ### Author Response · Authors · 2025-11-22
> **Continued Response**
>
> > It would be interesting to see how DrQ+BRO or DrQ+SimBa performs as compared to DrQ+XQC, considering that BRO and SimBa papers did not include vision-based DMC experiments. Do authors think that these methods would also perform worse than XQC in vision experiments?
>
> While indeed very interesting, we believe that running these experiments is out of scope and far exceeds our available compute budget. One vision-based humanoid run takes ~1 week on a single GPU.
>
> As a side note, after submission, we spoke with the SimbaV2 authors, and they informed us that they were struggling to scale SimbaV2 to vision-based tasks. This makes us believe that XQC significantly outperforms these baselines on vision-based tasks, which in the future needs to be confirmed, of course.
>
> ---
>
> > Could the authors summarize the shortcomings of their theoretical analysis? What are the differences between the assumptions and real-world problems?
>
> The goal of the theoretical analysis is to compare the impact of the different objectives (CE and MSE) on optimization to understand why the CE loss worked so much better for the same function approximator. To ensure the condition number was finite, we required the smallest absolute eigenvalue to be greater than zero in order to bound the largest possible condition number. This requires the neural network parameter Hessian to be positive semi-definite, which is why we assume weight decay ensures this. In practice, we found the neural network Hessian was sometimes negative definite (which is not surprising) but in practice the condition number did not explode as you can see in Figures 3 & 4.
> We have additionally added an ablation with weight decay to Figure 10 as well as loss-landscape conditioning results to Appendix I. As expected, weight decay has no significant effect on XQC.
>
> ---
>
> > It would be interesting to see how scaling improves conditioning on its own as compared to using bn + wn + categorical loss.
>
> We have added loss-landscape conditioning experiments of a dense network with 2, 4 and 6 layers to Appendix I. As expected, the dense architecture has bad conditioning compared to all the other investigated architectures. Scaling, at least for those numbers of layers, does not show a meaningful trend.
>
> ---
>
> > Batch normalization is interesting because the batch statistics depend on the sampled batch (and the variance of these statistics depends on the batch size). To this end, it would be interesting to see how robust XQC is to different batch size settings and whether the increased variance improves the optimization landscape.
>
> This is a very interesting point. We have added ablations for the different batch sizes to Figure 10 and loss-landscape conditioning experiments to Appendix I. We find XQC’s performance to be robust against changes in batch size, which has no significant influence on the performance or on the conditioning.
>
> ---
>
> > Nitpicks:
>
> We have fixed the typos, thank you for pointing them out!

---

> > ### Comment · Reviewer_ATvv · 2025-11-27
> >
> > Thank you for the responses. I have no further questions, I am raising my score to accept.

---

> > > ### Author Response · Authors · 2025-11-27
> > >
> > > Thank you for your constructive feedback during the rebuttal. We are happy that we could answer all your questions to your satisfaction, and thank you for raising your score.

---

### Official Review · Reviewer_dowu · 2025-10-30

**Soundness:** 2
**Presentation:** 3
**Contribution:** 2
**Rating:** 4
**Confidence:** 3

**Summary:**

This paper proproes XQC, studying how to make critic optimization in deep RL better conditioned by systematically revisiting three ingredients—normalization layers, weight normalization/projection, and the critic loss—and ties them together through a Hessian-based analysis. It shows that, contrary to common practice favoring LN, BN yields local loss landscapes with dramatically smaller condition numbers during learning; WN leverages scale invariance to stabilize the effective learning rate (ELR); and a categorical cross-entropy (CE) distributional loss induces a better-conditioned objective than MSE, with a bounded Hessian condition number under mild assumptions. Building on these findings, the authors propose XQC, a simple SAC extension that combines BN, WN, and a CE Bellman error, arguing the trio acts synergistically to improve conditioning and stabilize ELR, which in turn boosts sample efficiency. Extensive experiments on 55 proprioceptive and 15 vision control tasks report state-of-the-art results against larger, more complex baselines, supporting the claim that improving optimization fundamentals can outperform mere scaling.

**Strengths:**

**Principled theory that links design choices.** The paper gives a clear Hessian-based account of why BN (vs. LN), CE (vs. MSE), and WN jointly improve critic optimization—showing BN yields much smaller condition numbers, CE produces a better-conditioned objective, and WN stabilizes the effective learning rate; it even derives bounded eigenvalue ranges/condition numbers under mild assumptions.

**Thorough and careful empirics at scale.** Results span 70 tasks (55 proprioceptive + 15 vision), use solid aggregate metrics (IQM AUC with 90% SBCIs, 10 seeds), and consistently show SOTA sample efficiency—including fair pixel-based comparisons by re-using DRQ-V2 encoders/hyperparameters—and diagnostic analyses that tie gains to stabilized ELR and gradients.

**Weaknesses:**

**Limited originality**. Similar ideas have been proposed in previous works, like BatchNorm in CrossQ (ICLR 2024); Replacing MSE with CE loss in EfficientZero (Neurips 2021), Stop Regression (ICML2024), BRC (Neurips 2025). It seems that this paper just analyzed the same problem from another perspective but reached a similar conclusion.

**Limited scope**. As mentioned before, similar conclusions have been proposed in previous work, and this paper seems to be an ensemble of them. However, it would be interesting if the authors could expand the scope of this paper to include more settings, such as model-based RL, to investigate the general efficacy of the proposed architecture rather than constraining the scope to only the actor-critic and model-free settings.

**Questions:**

See above.

---

> ### Author Response · Authors · 2025-11-22
> **Response to Reviewer dowu**
>
> We thank the reviewer for their review and acknowledging the good presentation. We were especially happy that they acknowledge our “principled theory that links design choices” which “even derives bounded eigenvalues”, as well as our “thorough and careful empirics at scale” that “consistently show SOTA sample efficiency”, “fair pixel-based comparisons” and “diagnostic analysis”, “supporting the claim that improving optimization fundamentals can outperform mere scaling”.
>
> In the following we want to answer the reviewer’s mentioned weaknesses and argue for the originality and extensive scope of our work.
>
> ---
>
> > Limited originality. Similar ideas have been proposed in previous works, like BatchNorm in CrossQ (ICLR 2024); Replacing MSE with CE loss in EfficientZero (Neurips 2021), Stop Regression (ICML2024), BRC (Neurips 2025). It seems that this paper just analyzed the same problem from another perspective but reached a similar conclusion.
>
> We appreciate the reviewer’s knowledge of the prior work. However, we respectfully disagree that the combination of these components constitutes limited novelty. In research, novelty often comes from understanding why components interact to produce superior results, rather than just introducing new mechanics.
>
> We argue that our contribution is distinct from an "ensemble" in three critical ways:
>
> - **Optimization-First vs. Scale-First**: The current zeitgeist in RL (e.g., SimbaV2, BRO, BRC) pushes for massive model scaling as a means to achieve sample efficiency. Our work fundamentally challenges this "bigger is better" trend. To the best of our knowledge, this is the first work to leverage eigenspectrum analysis in investigating the conditioning of loss landscapes induced through different architectural choices in RL. We propose that the loss landscape is more critical to performance than architectures that scale to a higher number of model parameters. By improving the observed condition number of our loss landscape directly through BN+WN+CE, we achieve SOTA performance with a fraction of the compute and parameters of the baselines the reviewer mentions.
>
> - **Synergistic Theory**: While papers like CrossQ and BRC introduced individual components, they did not analyze the synergy that we uncover. Our Hessian eigenvalue analysis reveals that it is specifically the trio of BN (conditioning), WN (stable ELR), and CE (bounded gradient norms and Hessian eigenvalues) that alleviates the optimization difficulties of the critic loss. This theoretical justification—linking architectural choices to the loss landscape—is a novel contribution missing from prior empirical works.
>
> - **Simplicity as a Feature**: We view the simplicity of assembling known components as a strength, not a weakness. XQC provides a "minimal working example" of a well-conditioned critic, stripping away the complexity of residual blocks, massive width, and custom components (like Shift & Norm and LERP) used in recent baselines, offering the community a highly efficient and reproducible foundation.
>
> - **Scaling to vision-based continuous control tasks**: We extend our architecture to vision-based RL, significantly improving on DrQ-v2 (which CrossQ, BRO, BRC and SimbaV2 did not demonstrate).

---

> ### Author Response · Authors · 2025-11-22
> **Continued Response**
>
> > Limited scope. As mentioned before, similar conclusions have been proposed in previous work, and this paper seems to be an ensemble of them. However, it would be interesting if the authors could expand the scope of this paper to include more settings, such as model-based RL, to investigate the general efficacy of the proposed architecture rather than constraining the scope to only the actor-critic and model-free settings.
>
> We thank the reviewer for the interesting suggestion to expand our analysis to model-based RL. While we agree that investigating well-conditioned architectures in model-based settings is a promising direction for future research, we believe it falls outside the scope of this work for the following reasons:
>
> - **Focus on Critic Optimization**: Our theoretical and empirical analysis is specifically targeted at the optimization landscape of the critic in model-free actor-critic methods. Model-based RL introduces a fundamentally different set of optimization dynamics—primarily centered around learning the world model and planning horizons—which would require a separate, dedicated analysis to do it justice.
>
> - **Extensive Existing Scope**: As the reviewer kindly noted, our current evaluation is already conducted at a large scale, covering 70 diverse tasks across 5 benchmarks, spanning both proprioceptive and vision-based domains. To the best of our knowledge, this represents one of the most comprehensive evaluations of model-free algorithms in recent literature.
>
> - **Significance of Model-Free RL**: Model-free actor-critic methods remain the dominant paradigm for continuous control due to their asymptotic performance and simplicity. Establishing the principles of well-conditioned optimization within this setting is a significant and self-contained contribution.
>
> ---
>
> We hope the reviewer agrees that while model-based RL is an exciting next step, the depth and breadth of our current analysis within the model-free paradigm constitute a complete scientific contribution.

---

### Official Review · Reviewer_UPAo · 2025-11-01

**Soundness:** 3
**Presentation:** 4
**Contribution:** 4
**Rating:** 8
**Confidence:** 4

**Summary:**

This paper proposes XQC, a simple modification of SAC that targets the optimization landscape of the critic. The key design is the synergistic use of batch normalization at every MLP layer, weight normalization via post-update unit-sphere projection, and a categorical distributional critic trained with cross-entropy. The authors argue this trio yields (i) bounded gradients, (ii) a nearly constant effective learning rate, and (iii) substantially better Hessian conditioning than typical alternatives. Empirically, XQC reports state-of-the-art sample efficiency across 55 proprioceptive and 15 vision DMC tasks, using fewer parameters and lower stated FLOP/s than strong baselines.

**Strengths:**

- Originality: Reframes sample efficiency through loss-landscape conditioning with BN, WN, CE and a principled rationale.
- Quality: Extensive experiments (10 seeds in most main results), scaling studies (UTD/width/depth), and insightful plasticity metrics.
- Clarity: The paper is easy to follow, the role of each component is well articulated, figures are generally informative.
- Significance: If the compute/latency picture is corroborated, XQC can influence default choices for actor–critic architectures.

**Weaknesses:**

- Reporting parameters and FLOP/s is helpful, yet the current figures seem potentially misleading. For a fair comparison, training/inference speed should be addressed to complete the real-world efficiency story.
- For pixels, the authors state that XQC keeps the vision encoder unchanged for fairness. That isolates MLP effects, but it also under-tests the hypothesis: BN/WN/CE improvements could plausibly extend into the conv encoder. This limits the reported vision gains.
- Figure 7 highlights sample efficiency but 10M buffer size is quite unrealistic in various setups. A 1M buffer setup would make the point under practical budgets.

**Questions:**

- XQC’s MLPs do not include residual/skip connections, while some baselines do. Given residuals’ impact on conditioning, it would be useful to justify their omission and/or report a small ablation.
- The bounded condition number result for CE relies on weight decay to enforce positive definiteness, yet the best empirical runs often avoid weight decay. The paper acknowledges this; still, a short empirical probe on “with vs without WD” vs conditioning would tighten the loop.

---

> ### Author Response · Authors · 2025-11-22
> **Response to Reviewer UPAo**
>
> We thank the reviewer for their very positive and helpful review. We were especially happy about the excellent presentation and contribution scores. Thank you!
>
> In the following, we answer all of the reviewer’s questions in detail:
>
> > Reporting parameters and FLOP/s is helpful, yet the current figures seem potentially misleading. For a fair comparison, training/inference speed should be addressed to complete the real-world efficiency story.
>
> We have added two columns to Figure 9, with training FPS and evaluation FPS of the MuJoCo Humanoid measured on an RTX 4090 workstation. XQC has 60% higher training FPS than the next quickest baseline. In terms of evaluation FPS it is slightly slower than SimbaV2. However, we want to emphasize that in practice the difference in evaluation speed is negligible, since all baselines are well above 1000 FPS, which is enough for even high frequency control tasks.
> Originally, we focused on reporting network parameters and FLOPs/s, because these metrics are more objective, as they are hardware-independent. However, we agree that FPS is a valuable addition to the efficiency analysis, especially for practitioners.
>
> ---
>
> > For pixels, the authors state that XQC keeps the vision encoder unchanged for fairness. That isolates MLP effects, but it also under-tests the hypothesis: BN/WN/CE improvements could plausibly extend into the conv encoder. This limits the reported vision gains.
>
> We respectfully disagree about under-testing our hypothesis. Our loss-landscape conditioning analysis in Section 3 is based on MLP architectures with different combinations of normalization/WN/losses. We did not test how this extends to CNN encoder architectures. As such, when extending our results to vision-based RL, tested our hypothesis on the MLP component and reused the standard DrQ-V2 encoder for a fair comparison. Empirically our architectural changes yield significant performance gains on the vision-based tasks.
>
> While extending the analysis to the vision-encoder for RL is an interesting question, this lies outside our computational budget during the rebuttal, with a single humanoid seed running for ~1 week alone. We are planning to continue this investigation in future work.
>
> ---
>
> > Figure 7 highlights sample efficiency but 10M buffer size is quite unrealistic in various setups. A 1M buffer setup would make the point under practical budgets.
>
> This was a typo on our side. In fact, we used the standard DrQ-V2 buffer size of 1M for all our vision-based experiments. We have corrected this typo in the hyperparameters Table 2.
>
> ---
>
> > XQC’s MLPs do not include residual/skip connections, while some baselines do. Given residuals’ impact on conditioning, it would be useful to justify their omission and/or report a small ablation.
>
> The rationale for not including skip connections is that we did not start our investigation from the perspective of the baselines, but instead built the algorithm from our conditioning investigation and only added necessary components. Overall, we are attempting to keep the architecture as simple as possible while maximizing performance.
>
> We have, however, added an ablation with residual connections and different network depths to Figure 10. Further, we added an investigation of the loss-landscape conditioning of skip connections to Appendix I. We find that skip connections do not have a significant effect on XQC’s performance or loss-landscape conditioning.
>
> ---
>
> > The bounded condition number result for CE relies on weight decay to enforce positive definiteness, yet the best empirical runs often avoid weight decay. The paper acknowledges this; still, a short empirical probe on “with vs without WD” vs conditioning would tighten the loop.
>
> We have added the requested ablation to Figure 10 and additional conditioning results to Figure 25 in Appendix I. The weight decay hyperparameters was chosen based on the magnitude of the smallest negative Eigenvalue (in line with what the theory suggests). The weight decay does not show a significant influence on the performance of XQC, as such for simplicity, we can leave it out. The conditioning results lead to the same conclusion.

---

### Author Response · Authors · 2025-11-22
**General Response**

We thank all reviewers for their thoughtful feedback and the high-quality reviews they provided.
We are encouraged by the very positive reception of our work. Specifically, all reviewers mention our **principled analysis**, the **extensive and strong empirical evaluation** across 70 tasks, the **efficiency and simplicity** of our proposed method, and the **clarity and presentation** of our manuscript.

We have updated the manuscript with the requested ablations (FPS measurements, skip connections, batch size, weight decay, and more). We provide a general response to the common questions. Additionally, we provide detailed individual responses to all reviewers, addressing each of their questions.

## Common questions among reviewers:
- **Limited novelty concerns**: Reviewers `dowu` and `ATvv` note that different combinations of the individual components of XQC (BN, WN, CE) have previously been used in prior works. We argue that the novelty of XQC lies in **identifying their specific synergistic effect** on the optimization landscape and building a simple and efficient architecture based on those findings.
To the best of our knowledge, this is the first work to leverage eigenspectrum analysis in investigating the conditioning of loss landscapes induced through different architectural choices in RL. Our thorough investigation enables us to challenge the current “bigger is better” paradigm and present a novel, simpler, and significantly more efficient architecture than competing baselines. We offer a deep insight into understanding why different architectures outperform others. Lastly, reviewer `UPAo` specifically highlights the originality of our manuscript as an excellent contribution, and `ATvv` and `79k1` score our contributions as good.

- **Real-world compute efficiency (FPS vs. FLOP/s)**: In our initial submission, we focused on reporting compute efficiency in terms of FLOP/s, as a hardware-independent and objective performance measure. Reviewers `UPAo` and `79k1` correctly pointed out that real-world performance in terms of training/evaluation frames-per-second (FPS) are important metrics for practitioners. To that end, we have added FPS measurements to Figure 9. XQC achieves **~60% higher Training FPS** compared to SimbaV2, the next fastest baseline. This confirms that our parameter efficiency translates directly into faster training times. XQC achieves >1300 evaluation FPS. While slightly lower than SimbaV2 (due to a slightly larger policy network), this remains well above the requirement for even high-frequency control tasks.

- **Omission of Residual Connections**: Reviewers `UPAo` and `ATvv` asked why XQC omits residual/skip connections, which are standard in baselines like BRO and SimbaV2. We built XQC from the bottom up based on our loss-landscape conditioning analysis. Given that we were striving for a simple and small architecture, there was no need to include skip connections. For the rebuttal, we added an ablation with skip-connections to Figure 10. We found no significant performance differences when adding skip connections to XQC.

- **Additional Ablations**: We incorporated several additional ablations that reviewers were interested in. Some of these were:
  - Weight decay: While weight decay is useful for our theoretical proofs to ensure positive definiteness of the network’s Hessian, omitting it in practice does not harm performance or conditioning. The practical XQC loss landscape is sufficiently well-behaved that omitting weight decay does not harm performance or conditioning (Figures 4 and 10 and Appendix I). As requested by reviewers, `UPAo` and `ATvv`, we have added ablations benchmarking the addition of weight decay to Figure 10 and in Appendix I.
  - We have added additional loss landscape conditioning analyses for different ablations that reviewers requested in Appendix I.

We hope these new experiments and clarifications address the reviewers' concerns. We have uploaded the revised paper, which reflects these changes.

---

### Meta-Review · Area_Chair_inur · 2026-01-07

**Summary:**

This paper presents a systematic analysis of BN, WN, and CE loss for critic optimization  and evaluate their combination as a unified training strategy (i.e., XQC). Reviewers positively highlighted the new analytical insights relevant to current RL research, along with thorough evaluations using multiple seeds and extensive benchmarks. The strong performance of XQC was also recognized as a strength. At the same time, reviewers questioned the level of novelty and recommended further validation in practical settings, along with additional ablations on residual connections, weight decay, and learning curves.

**Reviewer Concerns:**

Several concerns from the reviewers were addressed. For example, the authors provided additional studies on weight decay, residual connections, and mean learning curves. The authors also extended the efficiency evaluation beyond FLOPs to FPS. Moreover, practical insights were offered with respect to real-world RL settings.

**Reviewer Scores:**

The interim average score was around 7. AC believes that minor clarifications would be helpful. These include further analysis of residual connections, which may leave additional room for improvement. In addition, extending the analysis to vision encoders could support a more comprehensive comparison. Reviewers consistently raised concerns about novelty; while the authors emphasized novel eigenspectrum analysis, XQC was viewed as an overlooked but not fundamentally new strategy. Overall, even if the discussion had continued enough, the scores would have likely remained at 7, leading to a poster acceptance decision.

---

### Decision · Program_Chairs · 2026-01-26

Accept (Poster)